# The Dual Nature of Plasticity Loss in Deep Continual Learning: Dissection and Mitigation

**Haoyu Albert Wang**[1,2,†]
haoyuwang18@fudan.edu.cn

**Wei P. Dai**[1,4,†,‡]
weidai@fudan.edu.cn

**Jiawei Zhang**[1,3]
Zhangjw22@m.fudan.edu.cn

**Jialun Ma**[1]
21307130025@m.fudan.edu.cn

**Mingyi Huang**[1,2]
myhuang20@fudan.edu.cn

**Yuguo Yu**[1,2,3,4,‡]
yuyuguo@fudan.edu.cn

1. Research Institute of Intelligent Complex Systems, Fudan University.
2. Institute of Science and Technology for Brain-Inspired Intelligence, Fudan University.
3. State Key Laboratory of Brain Function and Disorders and MOE Frontiers Center for Brain Science, Institutes of Brain Science, Fudan University.
4. Shanghai Artificial Intelligence Laboratory.
† These authors contributed equally to this work.
‡ Corresponding author.

## Abstract

Loss of plasticity (LoP) is the primary cause of cognitive decline in normal aging brains next to cell loss. Recent works show that similar LoP also plagues neural networks during deep continual learning (DCL). While it has been shown that random perturbations of learned weights can alleviate LoP, its underlying mechanisms remain insufficiently understood. Here we offer a unique view of LoP and dissect its mechanisms through the lenses of an innovative framework combining *the theory of neural collapse* and *finite-time Lyapunov exponents* (FTLE) analysis. We show that LoP actually consists of two contrasting types: (i) type-1 LoP is characterized by highly negative FTLEs, where the network is prevented from learning due to the collapse of representations; (ii) while type-2 LoP is characterized by excessively positive FTLEs, where the network can train well but the growingly chaotic behaviors reduce its test accuracy. Based on these understandings, we introduce *Generalized Mixup*, designed to relax the representation space for prolonged DCL and demonstrate its superior efficacy vs. existing methods.

## 1 Introduction

Loss of neural plasticity has been identified as the main reason for the progressive cognitive decline in normal aging brains without neurodegenerative diseases (i.e., not due to loss of neurons) [4]. Deep learning systems, despite their success across various domains, also struggle with a similar loss of plasticity (LoP) with widespread effects in deep reinforcement learning (DRL) [23, 1, 28, 20] and deep continual learning (DCL, e.g., Fig. 1a) [7]. Unlike catastrophic forgetting, which describes a neural network's poor performance on previous tasks after learning new tasks, LoP is characterized by the deteriorating performance on new tasks which eventually prevents the system from learning continually. Moreover, given the size of datasets and high energy demand of modern AI training [30], the cost of time and resource required to re-train the existing networks from scratch to adopt new tasks is increasing prohibitive [31]. Therefore, it is crucial that we understand and tame LoP as the need is higher than ever for adaptive AI that can efficiently learn new tasks continually .

39th Conference on Neural Information Processing Systems (NeurIPS 2025).

**Related Work**   The study of the declining test performance of a continually trained deep neural networks (DNN) began with the efforts to understand different phases of training [2] and incremental learning [5]. Later LoP gained its name, i.e. loss of plasticity, and the most attention within the community of DRL, as DRL networks are constantly facing shifting objectives and changing input distribution with the evolving policies [14, 16, 24, 23, 20, 21, 22, 12, 28, 7, 6, 1, 10]. Recent works also show that supervised DCL suffers from the same pathology [6, 7, 8, 17, 18, 22, 10].

Plenty of studies have attempted to explain LoP. One commonly cited cause is the accumulation of dormant or saturated neurons [28, 8, 7]. Activations such as ReLU and tanh are known to suffer from such effects where their outputs becomes permanently zero or constant, leading to vanishing gradients which reduce the network's expressivity. Another line of work attributes LoP to a decrease in feature rank [16, 7, 20], which characterize the drop in the diversity of internal representations as training progresses, weakening the network's ability to adapt to new tasks. Other hypotheses include the growth of weight norms [22], which may destabilize training, and the loss of gradient curvature[18], where training dynamics restrict gradient directions to a subspace, thereby constraining the exploration of alternatives. These studies provide valuable insights toward the understanding of LoP. However, some argue that LoP can still occur in absence of the aforementioned conditions [21, 12]. Therefore, a more comprehensive understanding of LoP is in need.

Various methods have also been proposed to mitigate LoP. They can be roughly categorized into reset-based, regularization-based, and architectural or optimizer-based approaches. Based on the observation that DNN exhibit the highest plasticity at initialization, an earlier work suggested retraining newly initialized networks on a replay buffer[14]. While effective, this strategy incurs significant computational cost. To reduce overhead, later works only apply resets on specific neurons instead, e.g., on the less "utilized" neurons [6, 7], on neurons with "zero firing rate" [10] or whose activation are relatively low within each layer[28]. Regularization-based methods are also motivated by the network's plasticity at initialization but take a softer approach, i.e., they encourage the network to retain some properties of its initial state. For instance, by applying an L2 regularization between the current weights and the initial weights (*L2-init*)[17], or with the Wasserstein-2 distance [18] for more proximity to the initialization. Similarly, distillation of initial features[20], as well as a "shrink and perturb" of the weights before each optimization[3] are proposed to maintain feature diversity throughout learning. Architectural and optimizer-based approaches are more intricate, including concatenating two ReLU activations to mitigate dormancy [1] (at the cost of increased model size) and applying layer normalization [21, 22]. Additionally, Lyle et al. [21] demonstrated that adjusting the decay rate of momentum or resetting the optimizer can help restore plasticity.

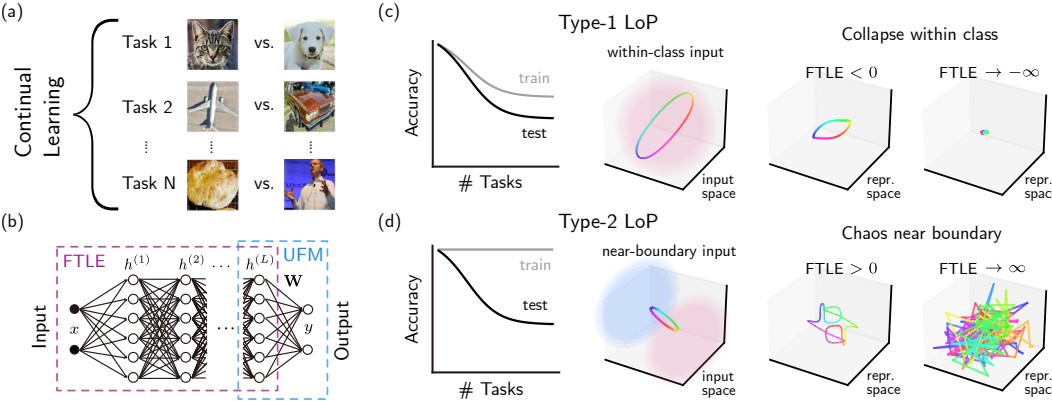

Figure 1:   **Illustration of LoP**, **(a)** Setup of "Continual ImageNet", adapted from [7]. **(b)** The effective framework to analyze LoP using FTLE and UFM. **(c)** and **(d)** Illustrations of the difference between type-1 and type-2 LoP on the level of task performance and their contrasting underlying causes, characterized by FTLE.

**Contribution**   As mentioned above, despite various approaches to understand and mitigate LoP, we are still not sure what drives LoP fundamentally. In this work, we comprehensively dissect and address LoP from a new perspective to provide deeper understandings along with a better treatment and less computational cost.

Specifically, (i) We offer an innovative framework combining finite-time Lyapunov exponent (FTLE) analysis [29] with the Unconstrained Feature Model (UFM) used in the theory of NC[36] (Fig 1b), where FTLE quantifies how representation space transforms during training and UFM provides the analytical tractability for optimization in representation space despite DNN's non-convexity. Using this framework, (ii) we identified the two types of LoP during DCL. On the level of task performance, they differ only in training accuracy, however, (iii) we further unveil that their underlying causes are exactly opposite: the collapsing of representation space within class (corresponding to increasingly negative FTLEs) is the direct cause of type-1 LoP, while chaotic behavior near the boundaries (corresponding to increasingly positive FTLEs) is the direct cause of type-2 LoP, (Fig 1c, d). (iv) Based on these understandings, we proposed and tested a generalized version of Mixup [35] as the better prescription for LoP.

## 2 Preliminaries

Here, we briefly list the preliminaries of our work, i.e., finite-time Lyapunov exponent (FTLE), the theory of Neural Collapse (NC) with unconstrained feature model (UFM), and set up the problem of LoP based on existing studies. Readers familiar with the subjects can safely skip this section.

**FTLE of DNN**  A DNN with $L$ hidden layers apply a composition of multiple nonlinear mappings, $\phi^{(\ell)}$, on input $x$ to its hidden layer representations $h^{(\ell)}, \forall \ell \in \{1 \dots L\}$ (Fig. 1b), and the penultimate layer representation reads:

$$h^{(L)} = \phi^{(L)} \circ \phi^{(L-1)} \circ \cdots \circ \phi^{(1)}(x), \tag{1}$$

The propagation of a local change in input, $\delta x$, is characterized by the network's Jacobian composed of derivatives of the activation $\mathbf{D}^{(\ell)}$ and weight matrices $\mathbf{W}^{(\ell)}$:

$$\delta h^{(\ell)} = \mathbf{D}^{(\ell)} \mathbf{W}^{(\ell)} \cdots \mathbf{D}^{(1)} \mathbf{W}^{(1)} \delta x^{(0)}, \tag{2}$$

This layer-by-layer mapping from the input space to its deep representation space can be interpreted as the evolution of a dynamical system, with layers as discrete time steps [29], therefore the network's sensitivity to local changes in input can be analogously described by the maximal Lyapunov exponent used for states $y(t)$ in dynamical systems, but with finite time, i.e., FTLE:

$$\lambda(y) = \lim_{t \to \infty} \frac{1}{t} \log \frac{|\delta y^{(t)}|}{|\delta y^{(0)}|} \implies \lambda^{(L)}(x) = \frac{1}{L} \log \frac{|\delta h^{(L)}|}{|\delta x^{(0)}|}. \tag{3}$$

By definition, a positive FTLE indicates significant local divergence in the representation, often corresponding to a classification boundary; and a negative FTLE suggests a convergence in the representation space for local inputs, usually belonging to the same class. Note that, *we use the unnormalized form of FTLE, $\lambda \equiv L\lambda^{(L)}$ in this paper.*

**Neural Collapse and the Unconstrained Feature Model**  Neural collapse (NC) describes a remarkable phenomenon observed during the terminal phase of training [25], where the network's last-layer features converge to the means within their class, and align with the classifier vectors optimally to form a simplex equiangular tight frame that enhance test performance even when the training error approaches zero.

Theoretical analysis of NC adopts the Unconstrained Feature Model (UFM), where the last-layer representations, $h_i^k$ for $i \in \{1 \dots n\}$, in each class $k$ are treated as free optimization variables just as the classifier weights $\mathbf{W}$ (Fig 1b), to circumvent the non-convexity of DNN, enabling analytical tractability[36, 34, 9]. Omitting the bias term in the last-layer output, the optimization of UFM is simply the sum of loss $\mathcal{L}$ over $K$ different classes against their labels, $y_k^{\text{target}}$ :

$$\min_{\mathbf{W}, H^k} \frac{1}{K} \sum_{k=1}^{K} \mathcal{L}(\mathbf{W} H^k, y_k^{\text{target}}), \tag{4}$$

where matrix $H^k \equiv [h_1^k, \dots, h_n^k]$. Prior studies on single classification tasks have demonstrated that the solution to this optimization inherently satisfies the properties of NC [36, 9, 27].

**Problem setup of LoP**   The training paradigm in DCL can be formally defined by considering a sequence of $T$ tasks $\{\mathcal{T}_1, \mathcal{T}_2, \ldots \mathcal{T}_T\}$, where each task $\mathcal{T}_t$ is associated with its own training set, $\mathcal{D}_{\text{train}}^{(t)}$, and test set, $\mathcal{D}_{\text{test}}^{(t)}$. The goal of DCL is to train a DNN that continually adapts to new tasks, updating the parameters of the network $\theta_t$ after each task $\mathcal{T}_t$ such that $\theta_t = \arg\min_\theta \mathcal{L}(\mathcal{D}_{(\text{test})}^{(t)}; \theta_{t-1})$, where $\mathcal{L}$ denotes the loss function [33, 11]. LoP then formally refers to the phenomena that the test performance of a network with continually updating parameters, $\theta_t$, becomes increasingly worse than that of a network trained from scratch with initialized parameters, $\theta_0$. In terms of loss, given a large enough number of trained tasks, $T'$, LoP occurs in the network when $\mathcal{L}(\mathcal{D}_{\text{test}}^{(t)}, \theta_0^*) < \mathcal{L}(\mathcal{D}_{\text{test}}^{(t)}, \theta_t)$ for all tasks $\mathcal{T}_t, \forall t > T'$, where $\theta_0^*$ denotes the parameters of a reinitialized network after training on task $\mathcal{T}_t$. Recently, Dohare et al. [7] has demonstrated the ubiquitous existence of LoP in DCL, particularly in the binary classification tasks where the network continually learns with consistent difficulty. Without loss of generality, this study focuses mostly on binary classification as well for ease of comparison, but we also tested our method on class-incremental tasks (Sec. 4).

## 3   Main Results

Although the theory of NC has successfully described the optimal representational structure trained on classification tasks, whether the theory fits the context of continual learning remains an open question. Moreover, the theory utilized UFM for analytical tractability, which ignores the mapping from input space to representation space altogether (Sec. 2). Thus, to dissect the underlying mechanisms of LoP, we develop a framework in Sec. 3.1 that combines the analysis of FTLE (which quantifies the geometric mapping between the input space and the representation space, see Sec. 2) with NC to provide the whole picture of training dynamics (Fig. 1b). Extending this framework to DCL, we identify two distinct types of LoP and their causes, which are verified with toy models in Sec 3.2 and 3.3. In Sec 3.4, we check the existence of two types of LoP for networks trained on ImageNet. Finally, in Sec 3.5, we introduce a generalized version of Mixup to address both types of LoP, grounded in our theoretical understanding of their underlying mechanisms.

### 3.1   Stretching and compressing Representation Space by Training

The gradient of cross-entropy (CE) loss with respect to the classifier weights and representations consists of a "push" term and a "pull" term [34]. This argument also holds for the mean squared error (MSE) loss.

Consider a binary classification task. Let $h_i^k \in \mathbb{R}^d$ be the penultimate layer $d$-dimensional representation of the $i$-th sample in class $k$, and let $\mathbf{w}_k \in \mathbb{R}^d$ denote the classifier weight vector of class $k$, where $k = A$ or $B$. The network output for paired samples can take the matrix form, $\mathbf{Y}_i = [\mathbf{w}_A^\top h_i^A, \mathbf{w}_B^\top h_i^A; \ \mathbf{w}_A^\top h_i^B, \mathbf{w}_B^\top h_i^B]$. Then, the corresponding one-hot labels of class $A$ and $B$ can be written in the corresponding matrix form, $\mathbf{Y}_i^{\text{target}} \equiv [1, 0; 0, 1]$. Thus, the MSE loss reads $\mathcal{L}_{\text{MSE}} = \frac{1}{2}\|\mathbf{Y} - \mathbf{Y}^{\text{target}}\|^2$. Taking its gradient over $\mathbf{w}_k$ for $k = A, B$, we have:

$$-\frac{\partial \mathcal{L}_{\text{MSE}}}{\partial \mathbf{w}_k} = \sum_i h_i^k \left(1 - \mathbf{w}_k^\top h_i^k\right) + \sum_i h_i^{k'} \left(0 - \mathbf{w}_k^\top h_i^{k'}\right), \tag{5}$$

where the first term pulls the weight vector $\mathbf{w}_k$ closer to the same-class representations $h_i^k$. The second term pushes it away from the representations $h_i^{k'}$ of class $k' \neq k$. Similarly, the gradient over $h_i^k$ shows two other "pull" and "push" terms acting on the representations:

$$-\frac{\partial \mathcal{L}_{\text{MSE}}}{\partial h_i^k} = \left(1 - \mathbf{w}_k^\top h_i^k\right) \mathbf{w}_k + \left(0 - \mathbf{w}_{k'}^\top h_i^k\right) \mathbf{w}_{k'}, \tag{6}$$

Altogether, they induce maximal class separability, from which we have the following theorem:

**Theorem 3.1.1 (Optimal Solution for Binary Classification)** *Under UFM and the assumption that both classes contain $n$ samples, $|A| = |B| = n$, consider the following optimization problem:*

$$\min_{\mathbf{W}, H} f(\mathbf{W}, H) := \frac{1}{2}\|\mathbf{W}H - Y\|_{\text{MSE}}^2 + \frac{\lambda_{\mathbf{W}}}{2}\|\mathbf{W}\|^2 + \frac{\lambda_H}{2n}\|H - H_0^\perp\|^2, \tag{7}$$

where $\lambda_{\mathbf{W}}$ and $\lambda_H$ are the coefficients of the two regularization terms that control the magnitude of the classifier weights, $\mathbf{W}$, and deviation of the learned representations $H$ from the initial representations $H_0^{\perp}$ orthogonal to $\mathbf{W}$. The optimal solution obeys:

$$\mathbf{w}_A^* = -\mathbf{w}_B^*, \quad h_i^{k,*} = \sqrt{\lambda_{\mathbf{W}}/\lambda_H}\, \mathbf{w}_k + h_{i,0}^{k,\perp}, \tag{8}$$

where $k = A$ or $B$, and $h_{i,0}^{k,\perp}$ is the $k$-component of $H_0^{\perp}$. (For proof, see Supplementary B.)

**Assumption 3.1.1** *Distribution of the real-world data satisfies the measure of concentration and the empty space phenomenon, two critical characteristics of high-dimensional data[32].*

We empirically verified that the datasets used in our experiments satisfy Assumption 3.1.1, (see Supplementary C for details). This validation allows us to directly employ FTLE analysis on local pairs of samples to quantify the transformation of the representational space, as the within-class regions and areas near classification boundaries are affected by local sample points only.

**Theorem 3.1.2 (FTLE in Representation Learning)** *The distance between two samples $i$ and $j$ from classes $k$ and $k'$ in representation space, $\Delta h_{ij}^{k,k'} \equiv h_i^k - h_j^{k'}$. can be quantified by FTLE, $\lambda_0$ and $\lambda_{trained}$ , each denotes its values before and after training, respectively.*

*(1) For samples from different classes ($k \neq k'$), training produces a positive FTLE ridge, stretching the space in-between, and the magnitude of stretch is captured by a stretching factor, $\alpha$:*

$$\log \alpha = \lambda_{\text{trained}}^{k,k'} - \lambda_0^{k,k'} > 0, \tag{9}$$

*(2) For samples of the same class ($k = k'$), training carves out a valley of negative FTLE, compressing the space in-between, and the degree of compression is given by a compressing factor, $\beta$:*

$$\log \beta = \lambda_{\text{trained}}^{k,k} - \lambda_0^{k,k} < 0. \tag{10}$$

(For proof, see Supplementary D.)

## 3.2 Type-1 LoP Induced by the Collapse of Representation Space

According to Theorem 3.1.2, within-class regions are compressed with a factor of $\beta_t < 1$ during the training of the same class samples. In the DCL setting, after learning $T$ tasks, the FTLE evolves as: $\lambda_{\text{trained},T}^{k,k} = \lambda_0^{k,k} + \log \prod_{t=1}^{T} \beta_t$. When the representational space is high-dimensional, the compression factor $\beta_t$ is close to 1; and $\beta_t$ decreases with lower effective dimensionality (for details, see Supplementary D). Over many tasks, the compressing effect accumulates:

$$\lambda_{\text{trained},T}^{k,k} \to -\infty \quad \text{as} \quad \prod_{t=1}^{T} \beta_t \to 0, \tag{11}$$

resulting in *the collapse of representation space*, where different samples (of the same class) become indistinguishable. This severely impairs the network's ability to learn new representations near the collapsed area for new tasks.

**Definition 3.2.1 (Type-1 LoP)** *In the settings of DCL, after trained by $T$ tasks, Type-1 LoP is resulted when the representations from different class, $h_i^k$ and $h_j^{k'}$ of a new task $\mathcal{T}_{T+1}$ come near the collapsed area, quantified by a highly negative FTLE $\lambda_{trained,T}^{k,k}$ (resulted from previous training). On the level of task performance, both training and test accuracy drop significantly, indicating a loss of capacity in learning.*

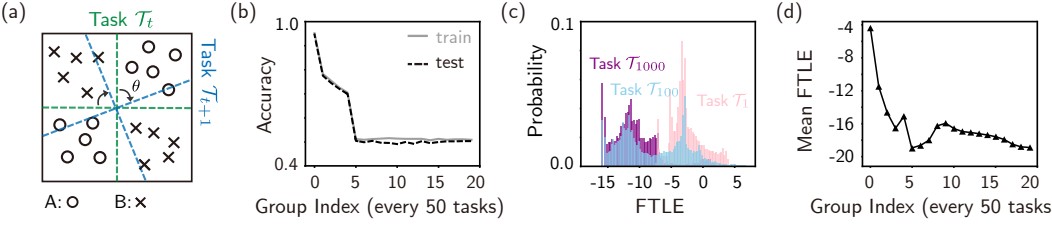

Figure 2: **Type-1 LoP in a low-dimensional minimalistic toy model**. **(a)** Illustration of the continual XOR classification task. **(b)** Training and test accuracy rapidly degrade over tasks. **(c)** Sampled distribution of FTLE shifts to lower values. **(d)** Mean FTLE are negative and decrease over tasks.

To intuitively understand Type-1 LoP, we construct a minimalistic DCL task based on XOR classification (Figure 2a). The inputs lie in a 2-D plane partitioned into four regions by two orthogonally crossing boundaries (diagonally opposite regions belong to the same class). After each task, the boundaries are rotated randomly by $\theta$, and the classification head is reinitialized. The random rotation simulates the lack of alignment between task boundaries typically observed in real-world scenarios, where consecutive tasks often exhibit a wide range of correlation. The network is a 12-layer MLP with 10 neurons per hidden layer, using *tanh* activation and MSE loss. As suggested by our theory, low-dimensional representational spaces are more susceptible to Type-1 LoP, making this architecture an ideal testbed.

The experimental results strongly support our analysis. As the number of tasks increases, both training and test accuracy degrade sharply and eventually to chance level (Fig. 2b). The sampled FTLE distribution (Fig. 2c) shifts toward highly negative values as the number of tasks increases, indicating a growing proportion of indistinguishable representations of inputs. The trend in mean FTLE (Figure 2d) further reveals the progressive compression of the representational space, driving the network to a state where it can no longer accommodate new task representations.

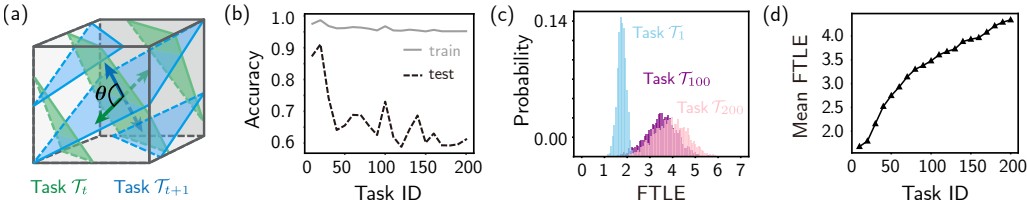

Figure 3: **Type-2 LoP in a high dimensional minimalistic toy model**. (a) Illustration of a 3-D projected 30-D toy model, where the hyperplanes are classification boundaries. (b) Training accuracy sustains near perfect while test accuracy degrades over time. (c) Sampled FTLE distribution quickly shifts toward larger values. (d) Mean FTLE are positive and increase over tasks.

## 3.3 Type-2 LoP Induced by the Over-stretched Boundaries and Chaotic Behaviors

According to Theorem 3.1.2, training a network with samples near the regions that cross the boundaries stretches the representation space with a factor of $\alpha_t > 1$. In the DCL setting, after learning $T$ tasks, the FTLE evolves as: $\lambda_{\text{trained},T}^{k,k'} = \lambda_0^{k,k'} + \log \prod_{t=1}^{T} \alpha_t$. This becomes more pronounced in high-dimensional representation spaces. Over many tasks, the accumulated effect yields:

$$\lambda_{\text{trained},T}^{k,k'} \to \infty \quad \text{as} \quad \prod_{t=1}^{T} \alpha_t \to \infty. \tag{12}$$

From the perspective of dynamical mean field theory in DNNs [26], large FTLE values signal the onset of chaotic dynamics. This manifests as extreme sensitivity to small input perturbations, where similar inputs are mapped to highly divergent representations. Despite the chaos, "uncollapsed" representation space support the network with enough expressive power to fit the training sets of the new tasks well. However, if the test samples come near the chaotic regions, the resulting representation becomes disordered and the network fails to generalize.

**Definition 3.3.1 (Type-2 LoP)** *In the setting of DCL, after trained by $T$ tasks, Type-2 LoP is resulted when the representation of a sample, $h_i^k$, from a new task $\mathcal{T}_{T+1}$ comes near a overly stretched area (resulted from previous training) quantified by a highly positive FTLE $\lambda_{\text{trained},T}^{k,k'}$. On the level of task performance, training accuracy is guaranteed by the highly expressive chaotic representation space while test accuracy degrades, indicating a loss of capacity in generalization.*

To empirically validate the emergence of Type-2 LoP, we designed a minimalistic toy model with a 30-D input space (a projection of which is illustrated in Fig. 3a) since Type-2 LoP is rooted in chaotic dynamics, as mentioned above. Data points are sampled uniformly from $[-1, 1]^{30}$. Three parallel hyperplanes partition the space into four regions with equal size, the same class assigned to every other region. For each task, the orientation of the hyperplane is chosen randomly and the classification head is reinitialized. The network is a 12-layer MLP with 300 neurons per hidden layer, using *tanh* activations and MSE loss. As predicted by our theory, representation spaces with higher dimensions are more susceptible to stretching, making this architecture an ideal testbed for Type-2 LoP.

The results clearly demonstrate Type-2 LoP. The mean of FTLE increases steadily with each task as well as its sampled distribution, indicating progressive expansion throughout the representation space (Fig. 3c and d). On the level of task performance, training accuracy sustains, while test accuracy quickly declines (Fig. 3b), revealing a growing generalization gap – the hallmark of Type-2 LoP.

## 3.4  Verifying LoP in Real Datasets

In Sec. 3.2 and 3.3, we constructed minimal synthetic toy models to illustrate two distinct types of LoP and validated our theoretical predictions. From the proof of theorem 3.1.2 we know that the dimensionality of representation space affects the rate of compression and expansion (see Supplementary D) but it does not necessarily prevent either one. In DCL, the prolonged training can gradually activate both mechanisms, leading to Type-1 LoP, Type-2 LoP, or both. The specific form depends on whether the representation space *where the new data points reside*, collapses or expands.

To validate our theory in a real-world setting, we evaluate LoP behaviors on the "Continual ImageNet" benchmark [7], a sequence of binary classification tasks designed for DCL (see details in Sec. 4). We use the same CNN setup from [7] for the high-dimensional case, and a narrowed version to model the low-dimensional case (see Supplementary E for details).

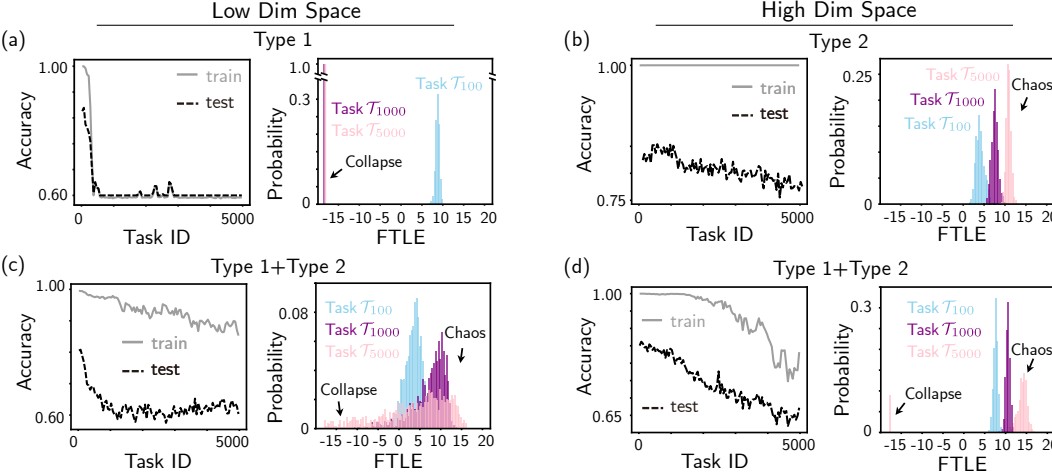

Figure 4: **Representative LoP scenarios in a real dataset. (a), (c)**: Low-dimensional representation space—(a) Type-1 LoP only; (c) Co-occurrence of Type-1 and Type-2 LoP. **(b), (d)**: High-dimensional representation space—(b) Type-2 LoP only; (d) Co-occurrence of both types.

Fig. 4 presents four representative cases that capture typical LoP behaviors with representation space of different dimensionality (D): In the low-D case of Fig. 4a, Type-1 LoP dominates, as indicated by a sharp drop in both training and test accuracy and a shift in FTLE distribution toward strongly negative values. In another low-D case (Fig. 4c), both types of LoP emerge. The FTLE distribution broadens significantly, with negative values reflecting collapse (Type-1) and large positive values reflecting chaotic expansion (Type-2). In the high-D case of Fig. 4b, only Type-2 LoP is observed. The FTLE distribution shifts toward positive values as network generalization degrades despite full training accuracy. In another high-D case (Fig. 4d), two types of LoP co-occur. Initially, Type-2 LoP dominates due to rapid stretching near boundaries. As more tasks are learned, within-class regions begin to collapse, confirmed by the bimodal expansion of the FTLE distribution.

## 3.5  Mitigating LoP with *Generalized Mixup*

Since we have shown that there are two types of LoP, an effective mitigation strategy must alleviate intra-class collapse and inter-class boundary expansion at the same time. Here, we draw inspiration from the classical *Mixup* [35]. It generates synthetic data by interpolating pairs of samples and their labels:

$$x^m = mx^K + (1-m)x^{K'}, \quad y^m = my^K + (1-m)y^{K'}, \tag{13}$$

where $m$ follows a Beta distribution. This formulation linearizes the decision boundaries and is effective against the excessive boundary stretching in Type-2 LoP. Specifically, chaotic behavior in high FTLE conditions, $|\delta h^{(L)}| = e^{\lambda^{(L)}(x)}|\delta x^{(0)}|$, let small input perturbations to be exponentially

amplified in the representation, often in the direction along the classifier $\mathbf{w}_K$. Classical *Mixup* encourages approximately linear transitions:

$$\mathbf{W}h(mx^K + (1-m)x^{K'}) \approx m\mathbf{W}h(x^K) + (1-m)\mathbf{W}h(x^{K'}), \tag{14}$$

thereby suppressing such instabilities and regularizing the representation near the decision boundary.

However, classical *Mixup* does not address Type-1 LoP, which is caused by representational collapse within a class. To mitigate this, we introduce *Generalized Mixup* (*G-mixup*), which modifies the label assignment during intra-class interpolation. For two samples $x_i^K$ and $x_j^K$ from the same class, we define:

$$x^m = mx_i^K + (1-m)x_j^K, y_K^m = y^K + \frac{M}{2} - M|0.5 - m|, \quad y_{K'}^m = y^{K'} - \frac{M}{2} + M|0.5 - m|, \tag{15}$$

in the case of binary classification, where $M$ is a confidence amplification factor. This encourages the network to expand the intra-class representation, countering collapse and maintaining variability within the class (Figure 5a). For example, if the original labels for `dog` and `cat` are $[0.8, 0.2]$ and $[0.2, 0.8]$, inter-class *Mixup* uses the standard convex interpolation (Fig 5 b): $y^m = m[0.8, 0.2] + (1-m)[0.2, 0.8]$, while intra-class *G-Mixup* (e.g., dog-dog mix) adjusts to (Fig 5 a): $y^m = [1 - 0.4|0.5 - m|, 0.4|0.5 - m|]$.

We provide PyTorch-like pseudo-code for implementing *G-Mixup* with both MSE and NLL losses in Supplementary F.

## 4 Experiments

We evaluate the effectiveness of *G-Mixup* on two continual learning benchmarks [7]: **Continual ImageNet** and **Class-Incremental CIFAR-100**. **Continual ImageNet** is a binary task-incremental setting with 5000 tasks in total, where the network is trained with MSE loss and a CNN backbone. This setup enables long-lifetime plasticity analysis. In contrast, **Class-Incremental CIFAR-100** uses CE loss and a ResNet-18[13] backbone under 20-task class-incremental learning, offering complementary insights into different continual learning scenarios.

In both benchmarks, we consider two baselines: (1) **Reinit**, which trains each task from scratch by leveraging the inherent plasticity of randomly re-initialized parameters, and (2) **No Intervention (No Intv.)**, a continual learning model trained without any LoP mitigation trick. Additionally, we compare *G-Mixup* with three representative LoP prevention methods across different strategies: regularization-based (**L2 Init** [17]), architecture-based (**LayerNorm** [21]), and reset-based (Continual Backpropagation, **CBP** [7]).

**Continual ImageNet** In this benchmark, the model sequentially learns to distinguish between 5000 binary classification tasks randomly sampled from the ImageNet dataset. Each class contains 600 training and 100 test images. Networks are trained for 250 epochs per task using SGD (batch size 100). Only classification head is reinitialized at the beginning of each task to simulate task-incremental learning.

We use a consistent architecture with three convolutional layers followed by three fully connected layers. We compare a wide network (matching [7]) representing high-dimensional space, with a narrower version for the low-dimensional case. Implementation details are provided in Supplementary G.

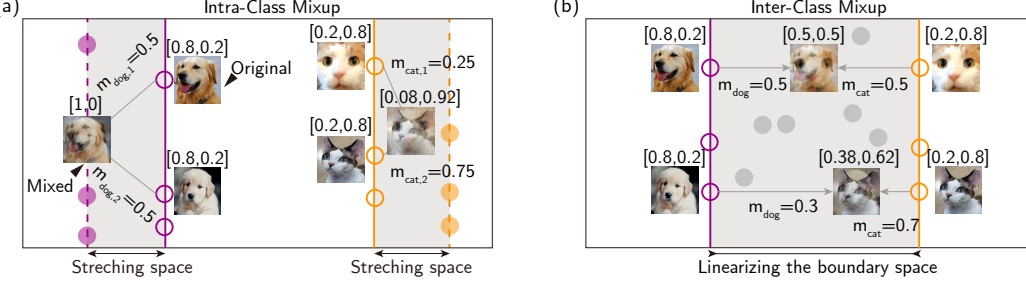

Figure 5: **Schematic of *Generalized Mixup* mitigating both type of LoP. (a)** Preventing intra-class representation collapse (Type-1). **(b)** Avoiding excessive inter-class boundary stretching (Type-2).

All models use ELU activation, except for CBP, which employs ReLU as originally proposed and achieves better performance with ReLU under the same setting. All results are averaged over five random seeds. Standard deviations are reported in Supplementary H.

Table 1: SmallConv Acc. on Continual ImageNet

| Task (×1000) | 0-1 | 1-2 | 2-3 | 3-4 | 4-5 |
|---|---|---|---|---|---|
| No Intv. | 0.817 | 0.805 | 0.562 | 0.500 | 0.500 |
| Retrained | 0.853 | 0.845 | 0.845 | 0.840 | 0.840 |
| L2 init | 0.804 | 0.796 | 0.786 | 0.785 | 0.788 |
| Layernorm | 0.753 | 0.760 | 0.759 | 0.751 | 0.751 |
| CBP | 0.834 | 0.847 | 0.846 | 0.847 | 0.857 |
| G-mixup[ours] | **0.866** | **0.881** | **0.885** | **0.880** | **0.879** |

Table 2: ConvNet Acc. on Continual ImageNet

| Task (×1000) | 0-1 | 1-2 | 2-3 | 3-4 | 4-5 |
|---|---|---|---|---|---|
| No Intv. | 0.794 | 0.778 | 0.604 | 0.537 | 0.500 |
| Retrained | 0.857 | 0.851 | 0.850 | 0.849 | 0.846 |
| L2 init | 0.814 | 0.805 | 0.800 | 0.803 | 0.807 |
| Layernorm | 0.782 | 0.768 | 0.752 | 0.749 | 0.755 |
| CBP | 0.848 | 0.867 | 0.864 | 0.863 | 0.878 |
| G-mixup[ours] | **0.875** | **0.896** | **0.899** | **0.894** | **0.896** |

Our findings show that while all methods mitigate LoP to some extent, *G-Mixup* consistently outperforms other methods across tasks and settings, demonstrating superior capacity to preserve plasticity over long task sequences.

We further analyze the FTLE distributions throughout training (see supplementary I). *G-Mixup* stabilizes FTLE within a moderate range, preventing both overly negative values (Type-1 LoP) and overly positive values (Type-2 LoP), confirming its role in controlling representation dynamics. Ablation studies (Supplementary J) validate that the benefits arise from G-Mixup itself, which effectively mitigates LoP, whereas classical Mixup severely suffers from representation collapse.

**Class-Incremental CIFAR-100**    In this setting, the network sequentially learns 5 new classes at each task, totaling 20 tasks to cover all 100 classes. Performance is evaluated on the full set of learned classes. Each task is trained for 200 epochs using SGD (batch size 100, weight decay 0.0005). The classification head remains fixed, with new heads added when new classes arrive, thereby simulating task-agnostic continual learning.

We adopt ResNet-18 as the backbone. We compare the standard ResNet-18 (high-dim) with a 0.25× scaled version (low-dim), where all channels and fully connected layers are reduced proportionally. We consider the setting of G-Mixup under a standard CE loss, which implicitly uses soft labels and may potentially improve model performance. To ensure fairness, all other methods are also trained with the same level of soft-label strength. All methods show noticeable performance gains after incorporating soft labels. Full configuration details are included in Supplementary K.

Table 3: 0.25x Resnet-18 Acc. on CIFAR100

| Task (×4) | 0-1 | 1-2 | 2-3 | 3-4 | 4-5 |
|---|---|---|---|---|---|
| No Intv. | 0.927 | 0.812 | 0.754 | 0.708 | 0.661 |
| Retrained | 0.926 | 0.800 | 0.745 | 0.702 | 0.666 |
| L2 init | 0.925 | 0.788 | 0.724 | 0.682 | 0.649 |
| Layernorm | 0.851 | 0.755 | 0.723 | 0.674 | 0.641 |
| CBP | 0.923 | 0.812 | 0.760 | 0.713 | 0.678 |
| G-mixup[ours] | **0.932** | **0.825** | **0.772** | **0.732** | **0.697** |

Table 4: Resnet-18 Acc. on CIFAR100

| Task (×4) | 0-1 | 1-2 | 2-3 | 3-4 | 4-5 |
|---|---|---|---|---|---|
| No Intv. | 0.907 | 0.847 | 0.812 | 0.778 | 0.743 |
| Retrained | 0.901 | 0.842 | 0.815 | 0.788 | 0.767 |
| L2 init | 0.922 | 0.829 | 0.785 | 0.752 | 0.723 |
| Layernorm | 0.847 | 0.782 | 0.763 | 0.728 | 0.697 |
| CBP | 0.907 | 0.847 | 0.816 | 0.788 | **0.768** |
| G-mixup[ours] | **0.928** | **0.864** | **0.832** | **0.800** | **0.768** |

As CIFAR-100 involves more complex multi-class predictions, task difficulty increases gradually and may exceed the capacity of narrower networks. Methods relying on random initialization or randomly sampled features require longer adaptation times in such settings. In contrast, *G-Mixup* facilitates more efficient learning and achieves higher accuracy across all tasks, even under low-capacity network configurations.

## 5    Conclusions

In this work, we present a unified theoretical framework that integrates FTLE analysis [29] with the UFM from the theory of Neural Collapse [36], offering a new perspective on the dynamics of representation space during continual learning. Through this framework, we reveal that LoP is composed of two contrasting types which arises from two distinct mechanisms: Type-1 LoP results from intra-class collapse of the representation space, marked by increasingly negative FTLEs, whereas Type-2 LoP stems from chaotic behavior near decision boundaries, indicated by increasingly positive FTLEs. Although both types can lead to test accuracy degradation, they differ in their geometric origins. Building on this insight, we propose *Generalized Mixup*, a method that explicitly

regularizes the geometry of the representation space to counteract both collapse and chaos. By jointly addressing the two types of LoP, *Generalized Mixup* consistently outperforms existing plasticity-preserving strategies while maintaining the computational efficiency. This work not only advances our understanding of plasticity loss through the lens of dynamical systems theory but also introduces a simple and effective approach for sustaining stable and adaptive continual learning.

**Relationship to Catastrophic Forgetting(CF)**   CF and LoP represent two complementary facets of the stability–plasticity dilemma in continual learning. CF reflects excessive plasticity, where new knowledge overwrites existing representations, whereas LoP denotes the opposite failure mode—a gradual loss of the ability to acquire new knowledge, even without explicit stability constraints. While CF has been extensively studied as a stability problem, our findings reveal that LoP emerges intrinsically from cumulative optimization and geometry distortion in the representation space. In contrast, stability-driven regularization methods such as EWC [15] or LwF[19] induce a different, externally imposed reduction in plasticity by design. Recognizing these dual failure modes provides a more complete picture of continual learning: sustaining long-term performance requires balancing memory retention against continual adaptability.

**Limitations**   While our proposed *Generalized Mixup* effectively mitigates both types of LoP in image classification tasks, it has several limitations. First, the method is specifically designed for supervised classification problems and may not generalize easily to other learning paradigms such as reinforcement learning, where the notion of label interpolation is less meaningful. Second, *Generalized Mixup* leverages the inherent label tolerance of image classification–namely, that the relative magnitudes of the output neurons are what determine the prediction, while the absolute values are largely irrelevant. This allows flexible interpolation of targets without significantly affecting performance. However, this assumption breaks down in tasks such as regression or structured prediction, where the output must precisely match the ground truth. Interpolating inputs and targets in such settings may introduce noise or bias, leading to degraded performance.

**Future Directions**   Directly regulating FTLE is a principled way to prevent LoP. However, estimating FTLE on real datasets is computationally expensive, so our direct-control experiments were verified only in toy settings. In this work, we mitigate LoP indirectly by constraining FTLE magnitudes via Generalized Mixup, which regularizes intra-class compression and inter-class boundary stretching without explicit Jacobian computation. A promising direction is to develop numerically efficient FTLE estimators and practical surrogates to enable online FTLE-aware training at scale.

**Acknowledgment**   We thank the support from the Science and Technology Innovation 2030 - Brain Science and Brain-Inspired Intelligence Project (2021ZD0201301), the National Natural Science Foundation of China (12201125,9257020), Shanghai Municipal Science and Technology (24JS2810400 and 21XD1400400). We thank Shanghai Institute for Mathematics and Interdisciplinary Sciences (SIMIS) for their financial support (SIMISID-2025-NC). The computations in this research were performed using the CFFF platform of Fudan University. Finally, we appreciate the discussions with Prof. Dongrui Wu at HUST.

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

# A List of Symbols Used in This Paper

| Symbol | Description |
| --- | --- |
| $h^{(\ell)}$ | Representation at layer $\ell$ of the network |
| $\phi^{(\ell)}$ | Nonlinear transformation at layer $\ell$ |
| $\mathbf{W}^{(\ell)}$ | Weight matrix at layer $\ell$ |
| $h^{(L)}$ | Penultimate-layer representation |
| $\delta x^{(0)}$ | Small perturbation in input |
| $\delta h^{(\ell)}$ | Perturbation at layer $\ell$ |
| $\lambda^{(L)}(x)$ | Finite-time Lyapunov exponent (FTLE) of input $x$ |
| $\lambda$ | Unnormalized FTLE, i.e., $\lambda = L\lambda^{(L)}$ |
| $h_i^k$ | Representation of sample $i$ in class $k$ |
| $\mathbf{w}_k$ | Classifier weight vector for class $k$ |
| $H^k$ | Matrix of representations for class $k$ |
| $\mathbf{W}$ | Classifier weight matrix |
| $Y$ | Label matrix for all samples |
| $\mathcal{L}$ | Loss function (typically MSE) |
| $\lambda_{\mathbf{W}}$ | Regularization coefficient for $\mathbf{W}$ |
| $\lambda_H$ | Regularization coefficient for $H$ |
| $H_0^\perp$ | Initial representation orthogonal to $\mathbf{W}$ |
| $\alpha$ | Stretching factor (inter-class expansion) |
| $\beta$ | Compression factor (intra-class collapse) |
| $\lambda_{\text{trained}}^{k,k'}$ | FTLE between class $k$ and $k'$ after training |
| $\lambda_0^{k,k'}$ | FTLE between class $k$ and $k'$ at initialization |
| $\mathcal{T}_t$ | $t$-th task in continual learning |
| $\theta_t$ | Network parameters after task $t$ |
| $\mathcal{D}_{\text{train}}^{(t)}$ | Training set of task $t$ |
| $\mathcal{D}_{\text{test}}^{(t)}$ | Test set of task $t$ |
| $m$ | Mixup interpolation coefficient |
| $x^m, y^m$ | Interpolated input and target in mixup |
| $M$ | Confidence amplification factor in Generalized Mixup |

# B Proof of Theorem 3.1.1

We prove Theorem 3.1.1 in main text that we restate as follows.

**Theorem B.1 (Optimal Solution for Binary Classification)** *Under UFM and the assumption that both classes contain $n$ samples, $|A| = |B| = n$, consider the following optimization problem:*

$$\min_{\mathbf{W},H} f(\mathbf{W}, H) := \frac{1}{2}\|\mathbf{W}H + \mathbf{b}\mathbf{1}^\top - Y\|_{\text{MSE}}^2 + \frac{\lambda_{\mathbf{W}}}{2}\|\mathbf{W}\|^2 + \frac{\lambda_H}{2n}\|H - H_0^\perp\|^2, \quad (16)$$

*where $\lambda_{\mathbf{W}}$ and $\lambda_H$ are the coefficients of the two regularization terms that control the magnitude of the classifier weights, $\mathbf{W}$, and deviation of the learned representations $H$ from the initial representations $H_0^\perp$ orthogonal to $\mathbf{W}$. The optimal solution obeys:*

$$\mathbf{w}_A^* = -\mathbf{w}_B^*, \quad h_i^{k,*} = \sqrt{\lambda_{\mathbf{W}}/\lambda_H}\,\mathbf{w}_k + h_{i,0}^{k,\perp}, \quad (17)$$

*where $k = A$ or $B$, and $h_{i,0}^{k,\perp}$ is the $k$-component of $H_0^\perp$.*

We begin by computing the gradients of the loss function with respect to $\mathbf{W}$ and $H$, and setting them to zero to obtain the optimality conditions.

The gradient of $\mathcal{L}$ with respect to $\mathbf{W}$ is given by

$$\nabla_{\mathbf{W}}\mathcal{L} = (\mathbf{W}H + \mathbf{b}\mathbf{1}^\top - Y)H^\top + \lambda_{\mathbf{W}}\mathbf{W} = 0. \quad (18)$$

The gradient with respect to $H$ is

$$\nabla_H \mathcal{L} = \mathbf{W}^\top (\mathbf{W}H + \mathbf{b}\mathbf{1}^\top - Y) + \frac{\lambda_H}{n}(H - H_0^\perp) = 0. \qquad (19)$$

To connect the two optimality conditions, we multiply both sides of Equation (18) on the left by $\mathbf{W}^\top$, yielding

$$\mathbf{W}^\top (\mathbf{W}H + \mathbf{b}\mathbf{1}^\top - Y)H^\top + \lambda_{\mathbf{W}} \mathbf{W}^\top \mathbf{W} = 0. \qquad (20)$$

Substituting Equation (19) into the first term of Equation (20) gives

$$-\frac{\lambda_H}{n}(H - H_0^\perp)H^\top + \lambda_{\mathbf{W}} \mathbf{W}^\top \mathbf{W} = 0. \qquad (21)$$

Rearranging terms, we obtain the following relationship between the optimal weight matrix $\mathbf{W}$ and the learned representation matrix $H$:

$$\lambda_{\mathbf{W}} \mathbf{W}^\top \mathbf{W} = \frac{\lambda_H}{n}(H - H_0^\perp)H^\top. \qquad (22)$$

Based on prior results from Neural Collapse theory [36], the optimal classification weight matrix converges to a Simplex Equiangular Tight Frame structure. In the binary classification case, this reduces to a simple antipodal configuration:

$$\mathbf{w}^A = -\mathbf{w}^B. \qquad (23)$$

Assuming the classification weights have reached their optimal configuration, the training dynamics of the representation $h_i^k$ under the MSE loss yield the following gradient:

$$-\frac{\partial \mathcal{L}_{\text{MSE}}}{\partial h_i^k} = \left(1 - \mathbf{w}_k^\top h_i^k\right)\mathbf{w}_k + \left(0 - \mathbf{w}_{k'}^\top h_i^k\right)\mathbf{w}_{k'}, \qquad (24)$$

where $k$ is the correct class and $k'$ is the incorrect class index.

This gradient lies entirely in the subspace spanned by $\mathbf{w}_k$ and $\mathbf{w}_{k'}$, implying that the updates to $h_i^k$ only occur in the plane defined by the classification weight vectors. Consequently, the component of $h_i^k$ orthogonal to this plane remains unchanged during training.

Thus, the representation $h_i^k$ can be decomposed as:

$$h_i^k = h_i^{\parallel} + h_{0,i}^\perp,$$

where $h_i^{\parallel} \in \text{span}\{\mathbf{w}_k, \mathbf{w}_{k'}\}$ and $h_{0,i}^\perp \perp \text{span}\{\mathbf{w}_k, \mathbf{w}_{k'}\}$. The term $h_{0,i}^\perp$ denotes the component of the initial representation that is orthogonal to the subspace spanned by the classification vectors $\mathbf{w}_k$ and $\mathbf{w}_{k'}$.

Then Equation 22 becomes:

$$\lambda_{\mathbf{W}} \mathbf{W}^\top \mathbf{W} = \frac{\lambda_H}{n} H^{\parallel} H^{\parallel,\top},$$

where $H_0^{\parallel}$ denotes the component of the initial representations that is parallel to the subspace spanned by the classification weight vectors.

This constraint provides a closed-form relationship between the classifier weights and the representation geometry at optimality.

For simplicity, we will denote $||\mathbf{W}||_F^2 = E$ and thus $||H^{\parallel}||_F^2 = \frac{n\lambda_{\mathbf{W}}}{\lambda_H} E$.

We now continue the proof by deriving a lower bound of the MSE loss. Our goal is to express the lower bound in terms of the Frobenius norm $E = ||\mathbf{W}||_F^2$.

Recall the MSE loss:

$$\mathcal{L}_{\text{MSE}} = \frac{1}{2}\sum_i \left|\left| \mathbf{W}h_i + \mathbf{b}\mathbf{1}^\top - y_i \right|\right|^2.$$

Since the orthogonal component $h_{0,i}^{\perp}$ remains unchanged with $\mathbf{W}h_{0,i}^{\perp} = 0$, we can express the MSE loss as:

$$\mathcal{L}_{\text{MSE}} = \frac{1}{2}\sum_i \left\| \mathbf{W}h_i^{\|} + \mathbf{b} - y_i \right\|^2.$$

To enable a tight application of the reverse triangle inequality, we re-center the target labels by defining:

$$b := \frac{1}{2}(y_A + y_B),$$

and rewrite the labels as:

$$\tilde{y}_i := y_i - b \in \{\pm (y_A - b)\}.$$

Now, the MSE loss becomes:

$$\mathcal{L}_{\text{MSE}} = \frac{1}{2}\sum_i \left\| \mathbf{W}h_i^{\|} - \tilde{y}_i \right\|^2.$$

We now apply the reverse triangle inequality:

$$\left\| \mathbf{W}h_i^{\|} - \tilde{y}_i \right\| \geq \left| \|\tilde{y}_i\| - \|\mathbf{W}h_i^{\|}\| \right|,$$

which becomes tight when $\mathbf{W}h_i^{\|} \parallel \tilde{y}_i$, and squaring both sides yields:

$$\left\| \mathbf{W}h_i^{\|} - \tilde{y}_i \right\|^2 \geq \left( \|\tilde{y}_i\| - \|\mathbf{W}h_i^{\|}\| \right)^2.$$

Next, since $h_i^{\|}$ is aligned with the classification direction $\mathbf{w}_k$, we can write (see a breif proof in Section B.1):

$$\|\mathbf{W}h_i^{\|}\| = \|\mathbf{W}\|_F \cdot \|h_i^{\|}\|.$$

Therefore, we obtain:

$$\left\| \mathbf{W}h_i^{\|} - \tilde{y}_i \right\|^2 \geq \left( \|\tilde{y}_i\| - \|\mathbf{W}\|_F \cdot \|h_i^{\|}\| \right)^2.$$

Substituting into the MSE loss gives:

$$\mathcal{L}_{\text{MSE}} \geq \frac{1}{2}\sum_i \left( \|\tilde{y}_i\| - \|\mathbf{W}\|_F \cdot \|h_i^{\|}\| \right)^2.$$

From earlier, we have $\|H^{\|}\|_F^2 = \sum_i \|h_i^{\|}\|^2 = \frac{n\lambda_{\mathbf{W}}}{\lambda_H}E$, where $E = \|\mathbf{W}\|_F^2$. Let $n$ denote the total number of samples. Define the average squared norm:

$$\frac{1}{n}\sum_i \|h_i^{\|}\|^2 = \frac{1}{n} \cdot \frac{n\lambda_{\mathbf{W}}}{\lambda_H}E.$$

Now apply Jensen's inequality to the convex function $f(x) = (b - a\sqrt{x})^2$ for $a > 0$, to get:

$$\frac{1}{n}\sum_i \left( \|\tilde{y}_i\| - \|\mathbf{W}\|_F \cdot \|h_i^{\|}\| \right)^2 \geq \left( \|\tilde{y}_i\| - \|\mathbf{W}\|_F \cdot \sqrt{\frac{1}{n}\sum_i \|h_i^{\|}\|^2} \right)^2.$$

Substitute the average squared norm:

$$\frac{1}{n}\sum_i \left( \|\tilde{y}_i\| - \|\mathbf{W}\|_F \cdot \|h_i^{\|}\| \right)^2 \geq \left( \|\tilde{y}_i\| - \|\mathbf{W}\|_F \cdot \sqrt{\frac{\lambda_{\mathbf{W}}}{\lambda_H}E} \right)^2.$$

Recall $\|\mathbf{W}\|_F = \sqrt{E}$, so we obtain:

$$\mathcal{L}_{\text{MSE}} \geq \frac{n}{2}\left( \|\tilde{y}_i\| - \sqrt{\frac{\lambda_{\mathbf{W}}}{\lambda_H}}E \right)^2.$$

Thus, the original optimization objective becomes:

$$\mathcal{L} = \mathcal{L}_{\text{MSE}} + \frac{\lambda_{\mathbf{W}}}{2}\|\mathbf{W}\|_F^2 + \frac{\lambda_H}{2n}\|H - H_0^\perp\|_F^2 \tag{25}$$

$$\geq \frac{n}{2}\left(\|\tilde{y}_i\| - \sqrt{\frac{\lambda_{\mathbf{W}}}{\lambda_H}} \cdot E\right)^2 + \frac{\lambda_{\mathbf{W}}}{2}E + \frac{\lambda_H}{2n} \cdot \frac{n\lambda_{\mathbf{W}}}{\lambda_H}E \tag{26}$$

$$= \frac{n}{2}\left(\|\tilde{y}_i\| - \sqrt{\frac{\lambda_{\mathbf{W}}}{\lambda_H}} \cdot E\right)^2 + \lambda_{\mathbf{W}}E. \tag{27}$$

This gives a clean lower bound entirely expressed as a function of $E = \|\mathbf{W}\|_F^2$. Since the loss function is continuous and coercive in $E$, the minimum is achieved at some finite $E^*$.

The conditions for the inequality to hold require representations $h_i^\parallel$ have equal norm (which can also be intuitively obtained from the symmetry conditions ) :

$$\|h_i^\parallel\| = \text{const}, \quad \forall i.$$

Also we have the following constrain equation 22:

$$\lambda_{\mathbf{W}}\|\mathbf{W}\|_F^2 = \frac{\lambda_H}{n}\|H^\parallel\|_F^2 \quad \Rightarrow \quad \frac{1}{n}\sum_i \|h_i^\parallel\|^2 = \frac{\lambda_{\mathbf{W}}}{\lambda_H}\|\mathbf{W}\|_F^2.$$

Therefore, we have:

$$\|h_i^\parallel\| = \sqrt{\frac{\lambda_{\mathbf{W}}}{\lambda_H}} \cdot \|\mathbf{w}_k\|, \quad \text{for all } i \text{ in class } k.$$

Thus the representation becomes:

$$h_i^{k,*} = \sqrt{\frac{\lambda_{\mathbf{W}}}{\lambda_H}}\,\mathbf{w}_k + h_{i,0}^{k,\perp},$$

where $h_{i,0}^{k,\perp} \in \text{span}(\mathbf{W})^\perp$ is the fixed orthogonal component inherited from the initial representation.

## B.1   A brief proof of $\|\mathbf{W}h_i^\parallel\| = \|\mathbf{W}\|_F \cdot \|h_i^\parallel\|$.

Under the neural collapse assumption 23 that the classification weight matrix is structured as

$$\mathbf{W} = \begin{bmatrix} \mathbf{w}_A^\top \\ -\mathbf{w}_A^\top \end{bmatrix},$$

and we define

$$h_i^\parallel := \tau_i \mathbf{w}_A, \quad \text{for some } \tau_i \in \mathbb{R},$$

to emphasize that the representation lies along the direction of $\mathbf{w}_A$, and $\tau_i$ is arbitrary.

We have:

$$\mathbf{W}h_i^\parallel = \tau_i\begin{bmatrix} \mathbf{w}_A^\top\mathbf{w}_A \\ -\mathbf{w}_A^\top\mathbf{w}_A \end{bmatrix} = \tau_i\|\mathbf{w}_A\|^2\begin{bmatrix} 1 \\ -1 \end{bmatrix}.$$

Thus, the squared norm becomes:

$$\|\mathbf{W}h_i^\parallel\|^2 = 2\tau_i^2\|\mathbf{w}_A\|^4.$$

On the other hand:

$$\|h_i^\parallel\| = |\tau_i| \cdot \|\mathbf{w}_A\|, \quad \Rightarrow \quad \|\mathbf{W}\|_F^2 = \|\mathbf{w}_A\|^2 + \|-\mathbf{w}_A\|^2 = 2\|\mathbf{w}_A\|^2.$$

Therefore,

$$\|\mathbf{W}h_i^\parallel\| = \sqrt{2} \cdot |\tau_i| \cdot \|\mathbf{w}_A\|^2 = \sqrt{2} \cdot \|\mathbf{w}_A\| \cdot |\tau_i| \cdot \|\mathbf{w}_A\| = \|\mathbf{W}\|_F \cdot \|h_i^\parallel\|.$$

This confirms that the equality

$$\|\mathbf{W}h_i^\parallel\| = \|\mathbf{W}\|_F \cdot \|h_i^\parallel\|$$

holds exactly under this structural assumption.

# C   Verification of High-Dimensional Data Properties in ImageNet

In the main text, we assume that real-world data distributions exhibit two critical characteristics of high-dimensional spaces: the *measure of concentration* and the *empty space phenomenon*. To verify these assumptions, we conducted the following experiments on the ImageNet dataset:

1. Measure of Concentration: We calculated the average pairwise distance between samples in the representation space, along with the variance of these distances. This analysis helps to verify whether the distances concentrate around their mean, as expected in high-dimensional spaces.

2. Empty Space Phenomenon: To assess this property, we randomly sampled 1000 points in the representation space and computed the minimum distance between each sample point and these randomly generated points. This test evaluates the relative sparsity of data in high-dimensional spaces.

The results, shown in figure 6, confirm that the ImageNet dataset adheres to both the *measure of concentration* and the *empty space phenomenon*. These findings validate our assumption about the high-dimensional nature of real-world data distributions.

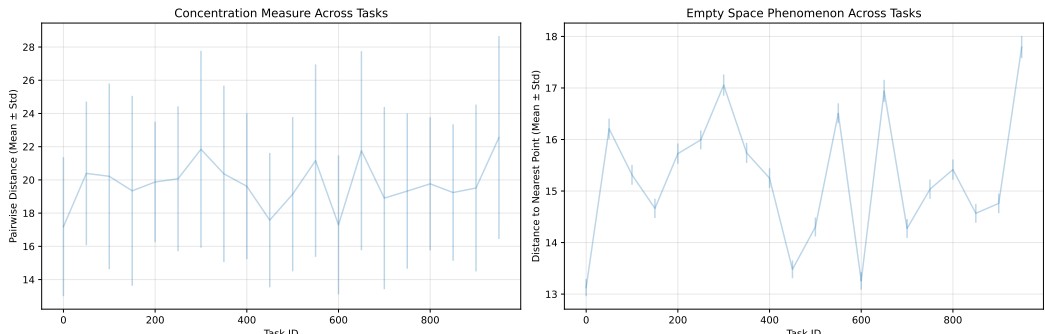

Figure 6: Verification of High-Dimensional Data Properties in ImageNet.

# D   Proof of Theorem 3.1.2

We prove Theorem 3.1.2 from the main text that we restate as follows.

**Theorem D.1 (FTLE in Representation Learning)** *The distance between two samples $i$ and $j$ from classes $k$ and $k'$ in representation space, $\Delta h_{ij}^{k,k'} \equiv h_i^k - h_j^{k'}$. can be quantified by FTLE, $\lambda_0$ and $\lambda_{trained}$, each denotes its values before and after training, respectively.*

*(1) For samples from different classes ($k \neq k'$), training produces a positive FTLE ridge, stretching the space in-between, and the magnitude of stretch is captured by a stretching factor, $\alpha$:*

$$\log \alpha = \lambda_{\text{trained}}^{k,k'} - \lambda_0^{k,k'} > 0, \tag{28}$$

*(2) For samples of the same class ($k = k'$), training carves out a valley of negative FTLE, compressing the space in-between, and the degree of compression is given by a compressing factor, $\beta$:*

$$\log \beta = \lambda_{\text{trained}}^{k,k} - \lambda_0^{k,k} < 0. \tag{29}$$

According to Theorem 3.1.1, the optimal representations satisfy:

$$h_i^{k,*} = \sqrt{\frac{\lambda_{\mathbf{W}}}{\lambda_H}}\, \mathbf{w}_k + h_{i,0}^{k,\perp},$$

where the trainable component lies entirely in the classification plane spanned by $\{\mathbf{w}_A, \mathbf{w}_B\}$, and the orthogonal component $h_{i,0}^{k,\perp}$ remains unchanged from initialization.

The trained representations become:

$$h_i^{A,*} = \sqrt{\frac{\lambda_{\mathbf{W}}}{\lambda_H}} \mathbf{w}_A + h_{i,0}^{A,\perp}, \quad h_j^{B,*} = \sqrt{\frac{\lambda_{\mathbf{W}}}{\lambda_H}} \mathbf{w}_B + h_{j,0}^{B,\perp}.$$

Hence, the inter-class representational difference after training becomes:

$$\Delta_{trained} h_{ij}^{A,B} = h_i^{A,*} - h_j^{B,*} = \left( h_{i,0}^{A,\perp} - h_{j,0}^{B,\perp} \right) + \sqrt{\frac{\lambda_{\mathbf{W}}}{\lambda_H}} \left( \mathbf{w}_A - \mathbf{w}_B \right).$$

In contrast, the inter-class representational difference before training is given by:

$$\Delta_0 h_{ij}^{A,B} = h_{i,0}^{A,\perp} - h_{j,0}^{B,\perp} + \left\langle h_{i,0}^A, \frac{\mathbf{w}_A}{\|\mathbf{w}_A\|} \right\rangle \frac{\mathbf{w}_A}{\|\mathbf{w}_A\|} - \left\langle h_{j,0}^B, \frac{\mathbf{w}_A}{\|\mathbf{w}_A\|} \right\rangle \frac{\mathbf{w}_A}{\|\mathbf{w}_A\|},$$

where the initial representations are projected onto the direction of $\mathbf{w}_A$.

The change in inter-class representational distance due to training satisfies:

$$\left\| \Delta_{\text{trained}} h_{ij}^{A,B} \right\|^2 - \left\| \Delta_0 h_{ij}^{A,B} \right\|^2 \tag{30}$$

$$= \frac{\lambda_{\mathbf{W}}}{\lambda_H} \cdot \|\mathbf{w}_A - \mathbf{w}_B\|^2 - \left\| \left\langle h_{i,0}^A, \frac{\mathbf{w}_A}{\|\mathbf{w}_A\|} \right\rangle \frac{\mathbf{w}_A}{\|\mathbf{w}_A\|} - \left\langle h_{j,0}^B, \frac{\mathbf{w}_A}{\|\mathbf{w}_A\|} \right\rangle \frac{\mathbf{w}_A}{\|\mathbf{w}_A\|} \right\|^2 \tag{31}$$

$$= \left( 4 \frac{\lambda_{\mathbf{W}}}{\lambda_H} - \left| \frac{\langle h_{i,0}^A - h_{j,0}^B, \mathbf{w}_A \rangle}{\|\mathbf{w}_A\|^2} \right|^2 \right) \cdot \|\mathbf{w}_A\|^2 \tag{32}$$

$$= \left( 4 \frac{\lambda_{\mathbf{W}}}{\lambda_H} \|\mathbf{w}_A\|^2 - \left| \left\langle h_{i,0}^A - h_{j,0}^B, \frac{\mathbf{w}_A}{\|\mathbf{w}_A\|} \right\rangle \right|^2 \right) \cdot \left\| \frac{\mathbf{w}_A}{\|\mathbf{w}_A\|} \right\|^2. \tag{33}$$

Similarly, the change in intra-class representational distance due to training is given by:

$$\left\| \Delta_{trained} h_{i,j}^{A,A} \right\|^2 - \left\| \Delta_0 h_{i,j}^{A,A} \right\|^2 \tag{34}$$

$$= \left\| h_i^{A,*} - h_j^{A,*} \right\|^2 - \left\| h_{i,0}^A - h_{j,0}^A \right\|^2 \tag{35}$$

$$= \left\| \left( h_{i,0}^{A,\perp} - h_{j,0}^{A,\perp} \right) + \sqrt{\frac{\lambda_{\mathbf{W}}}{\lambda_H}} \left( \mathbf{w}_A - \mathbf{w}_A \right) \right\|^2 \tag{36}$$

$$- \left\| \left( h_{i,0}^{A,\perp} - h_{j,0}^{A,\perp} \right) + \left( \left\langle h_{i,0}^A, \frac{\mathbf{w}_A}{\|\mathbf{w}_A\|} \right\rangle - \left\langle h_{j,0}^A, \frac{\mathbf{w}_A}{\|\mathbf{w}_A\|} \right\rangle \right) \frac{\mathbf{w}_A}{\|\mathbf{w}_A\|} \right\|^2 \tag{37}$$

$$= - \left\| \left( \left\langle h_{i,0}^A - h_{j,0}^A, \frac{\mathbf{w}_A}{\|\mathbf{w}_A\|} \right\rangle \right) \frac{\mathbf{w}_A}{\|\mathbf{w}_A\|} \right\|^2 \tag{38}$$

$$= - \left\| \left( \frac{\langle h_{i,0}^A - h_{j,0}^A, \mathbf{w}_A \rangle}{\|\mathbf{w}_A\|^2} \right) \mathbf{w}_A \right\|^2 \tag{39}$$

$$= - \left\| \left\langle h_{i,0}^A - h_{j,0}^A, \frac{\mathbf{w}_A}{\|\mathbf{w}_A\|} \right\rangle \frac{\mathbf{w}_A}{\|\mathbf{w}_A\|} \right\|^2 \tag{40}$$

$$\tag{41}$$

Thus, the key is to evaluate the following projection term:

$$\left| \left\langle h_{i,0}^K - h_{j,0}^K, \frac{\mathbf{w}_A}{\|\mathbf{w}_A\|} \right\rangle \right|^2.$$

This can be decomposed as:

$$\left| \left\langle h_{i,0}^K, \frac{\mathbf{w}_A}{\|\mathbf{w}_A\|} \right\rangle - \left\langle h_{j,0}^K, \frac{\mathbf{w}_A}{\|\mathbf{w}_A\|} \right\rangle \right|^2 .$$

Note that this expression captures the difference in the initial projections of the representations $h_{i,0}^K$ and $h_{j,0}^K$ onto the normalized classification direction $\mathbf{w}_A/\|\mathbf{w}_A\|$. .

Under this assumption, the original expression reduces to a geometric comparison between three random unit vectors in high-dimensional space: two initial representations $z_1, z_2$, and one classification direction $w$. Let us define:

$$\cos \theta_1 = \cos\left(\angle(z_1, w)\right), \quad \cos \theta_2 = \cos\left(\angle(z_2, w)\right).$$

We are interested in the expected deviation:

$$z = |\cos \theta_1 - \cos \theta_2|^2 .$$

According to Lemma D.1, the distribution of $\cos \theta$ between two independent unit vectors in $d$-dimensional Euclidean space follows the density:

$$p(\cos \theta) = \frac{\Gamma\left(\frac{d}{2}\right)}{\Gamma\left(\frac{d-1}{2}\right)\sqrt{\pi}} (1 - \cos^2 \theta)^{(d-3)/2}.$$

Let $\cos \theta_1 \sim p(x)$, $\cos \theta_2 \sim p(y)$. The distribution of their absolute difference $z = |x - y|^2$ can be written as:

$$p_z(z) = \int_{-1}^{1} \int_{-1}^{1} p(x)p(y)\, \delta(z - |x - y|^2)\, dx\, dy.$$

By symmetry, this simplifies to:

$$p_z(z) = \frac{C^2}{\sqrt{z}} \int_{-1}^{1-\sqrt{z}} (1 - x^2)^{(d-3)/2} \cdot \left(1 - (x + \sqrt{z})^2\right)^{(d-3)/2} dx,$$

where $C$ is the normalization constant ensuring $\int_{-1}^{1} p(x)\, dx = 1$.

Although the exact form of $p_z(z)$ is analytically intractable, it can be efficiently evaluated numerically. In the following figure, we plot the cumulative distribution function (CDF) of $z$. Here, the dimension $d$ refers to the width of the representation space, i.e., the dimensionality of the representation $h \in \mathbb{R}^d$. As $d$ increases, the distribution becomes increasingly concentrated near zero. Conversely, when the dimension $d$ is small, the distribution of $z$ becomes more dispersed and exhibits heavier mass at larger nonzero values.

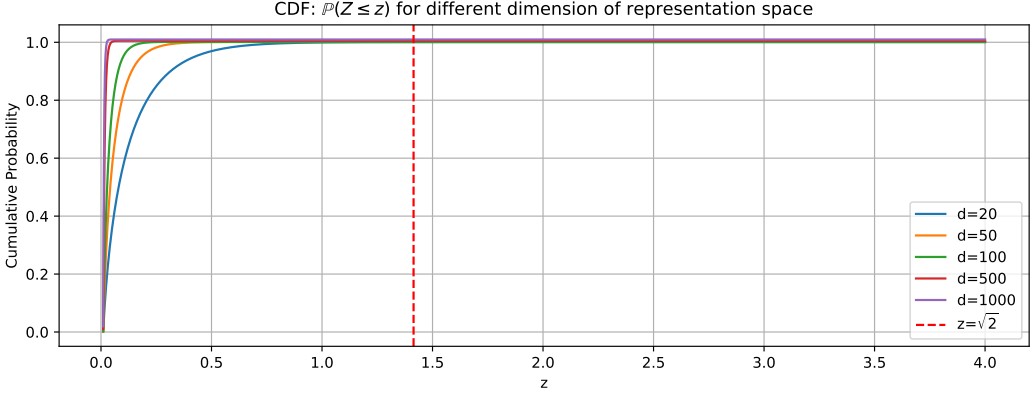

Figure 7: Numerical solution of the CDF of $z$.

**Within-Class Representation Collapse.** For the within-class region, in Eq 40, we have already shown that training changes the pairwise distance between representations as:

$$\left\|\Delta_{\text{trained}}h_{ij}^{K,K}\right\|^2 - \left\|\Delta_0 h_{ij}^{K,K}\right\|^2 = -\left\|\left\langle h_{i,0}^A - h_{j,0}^A, \frac{\mathbf{w}_A}{\|\mathbf{w}_A\|}\right\rangle \frac{\mathbf{w}_A}{\|\mathbf{w}_A\|}\right\|^2.$$

As we have analyzed earlier, the scalar projection term

$$z = \left\langle h_{i,0}^A - h_{j,0}^A, \frac{\mathbf{w}_A}{\|\mathbf{w}_A\|}\right\rangle$$

follows a distribution that becomes increasingly concentrated near zero as the representation dimensionality $d$ grows, and conversely, spreads more widely when $d$ is small.

Define the within-class shrinkage factor as:

$$\|\Delta_{\text{trained}}h_{ij}^{K,K}\| = \beta \cdot \|\Delta_0 h_{ij}^{K,K}\|, \quad \text{with } \beta < 1.$$

Then the FTLE after training is given by:

$$\lambda_{\text{trained}}^{K,K}(x) = \lambda_0^{K,K}(x) + \log \beta(x).$$

This implies that the FTLE will slightly decrease or remain nearly unchanged $\beta \lesssim 1$, when $d$ is large, but may shrink significantly when $d$ is small, due to larger projection fluctuations.

Importantly, while this within-task shrinkage effect may appear mild for a single task, in the continual learning setting it can accumulate over time, especially when repeated over multiple tasks. This accumulated shrinkage can lead to a significant compression of the local representation space, i.e., representation space collapse. This analysis is consistent with our empirical observations: in low-dimensional settings, collapse occurs reliably even for a small number of tasks, while in higher-dimensional spaces, collapse may or may not occur depending on the specific training setup and number of tasks.

**Boundary Space Over-Stretched.** For the inter-class region, Eq. 40 shows that training modifies the pairwise distance between representations of different classes as:

$$\left\|\Delta_{\text{trained}}h_{ij}^{A,B}\right\|^2 - \left\|\Delta_0 h_{ij}^{A,B}\right\|^2 = \left(4 \cdot \frac{\lambda_{\mathbf{W}}}{\lambda_H}\|\mathbf{w}_A\|^2 - \left|\left\langle h_{i,0}^A - h_{j,0}^B, \frac{\mathbf{w}_A}{\|\mathbf{w}_A\|}\right\rangle\right|^2\right).$$

To estimate the first term, we recall from the optimization analysis of Equation 27 in Section D that the total loss satisfies:

$$\mathcal{L}(E) \geq \frac{n}{2}\left(\|\tilde{y}_i\| - \sqrt{\frac{\lambda_{\mathbf{W}}}{\lambda_H}} \cdot E\right)^2 + \lambda_{\mathbf{W}}E,$$

where the minimum occurs approximately at

$$E^* = \sqrt{\frac{\lambda_H}{\lambda_{\mathbf{W}}}} \cdot \|\tilde{y}_i\| - \frac{\lambda_H}{n} \approx \sqrt{\frac{\lambda_H}{\lambda_{\mathbf{W}}}} \cdot \|\tilde{y}_i\|.$$

This is because $n$ is typically large compared to $\lambda_H$, and $\lambda_{\mathbf{W}} \sim \lambda_H$ in magnitude. Hence,

$$\|\mathbf{w}_A\|^2 \approx \frac{1}{2} \cdot \sqrt{\frac{\lambda_H}{\lambda_{\mathbf{W}}}} \cdot \|\tilde{y}_i\| \quad \Rightarrow \quad 4 \cdot \frac{\lambda_{\mathbf{W}}}{\lambda_H}\|\mathbf{w}_A\|^2 \approx \sqrt{2}.$$

In Figure 7, we overlay this estimate (red dashed line) on the empirical CDF of the squared projection term

$$z = \left|\left\langle h_{i,0}^A - h_{j,0}^B, \frac{\mathbf{w}_A}{\|\mathbf{w}_A\|}\right\rangle\right|^2,$$

we observe that in both low- and high-dimensional spaces, most values of $z$ fall below this estimated threshold. This indicates a net expansion of the representation space between classes during training.

Moreover, the distribution of the projection term $z$ becomes more concentrated around zero as the representation dimensionality $d$ increases, while it spreads more widely when $d$ is small. Therefore, although expansion dominates across dimensions, it is more significant in high-dimensional spaces, where the lower baseline projection leads to faster growth.

We formalize this expansion using a stretch factor:

$$\|\Delta_{\text{trained}} h_{ij}^{A,B}\| = \alpha \cdot \|\Delta_0 h_{ij}^{A,B}\|, \quad \alpha > 1.$$

The resulting FTLE becomes:

$$\lambda_{\text{trained}}^{K,K'}(x) = \lambda_0^{K,K'}(x) + \log \alpha(x),$$

indicating a positive FTLE ridge across inter-class boundaries, and reflecting a local "stretching effect" that enhances class separability.

While the effect of class boundary expansion is immediately pronounced in high-dimensional settings, its manifestation in low-dimensional networks can be subtle within a single task. However, under continual learning, where multiple tasks accumulate over time, the expansion effect may progressively intensify and eventually give rise to chaotic behavior. Consequently, we consistently observe clear boundary stretching in high-dimensional models, whereas in low-dimensional models, such dynamics may emerge only after sufficient task accumulation. This observation aligns with our findings reported in the main text.

**Lemma D.1** *In high-dimensional spaces, the angle $\theta$ between two randomly sampled vectors $x, y \in \mathbb{R}^d$ follows the distribution:*

$$p_d(\theta) = \frac{\Gamma\left(\frac{d}{2}\right)}{\Gamma\left(\frac{d-1}{2}\right)\sqrt{\pi}} \sin^{d-2}\theta,$$

*where $\Gamma(\cdot)$ is the gamma function[32].*

# E   Experimental Setup for Verifying LoP in Real-World Datasets

To validate our discovery of two different types of LoP across realistic datasets, we use the experimental setup described in Supplementary H, using identical architectures, training protocols, and hyperparameters. Across all settings, we consistently observe the two types of LoP as well as hybrid cases where both types coexist, differing only in their onset speed and severity.

In the main text Fig. 4, we visualize model performance under four representative single-run configurations :

- **(a)** ReLU + CE with the architecture in Table 5
- **(b)** Tanh + MSE with the architecture in Table 6
- **(c)** Tanh + MSE with the architecture in Table 5
- **(d)** ReLU + CE with the architecture in Table 6

All other experimental conditions remain identical across these runs.

# F   Generalized Mixup Pseudo Code

*Generalized Mixup* (*G-Mixup*) is designed as a simple, plug-and-play module that can be seamlessly integrated into existing training pipelines. It minimally modifies the classical *Mixup* [35] by adapting the label interpolation rule depending on whether the two samples belong to the same class (intra-class) or different classes (inter-class). It supports both Mean Squared Error (MSE) and Negative Log-Likelihood (NLL) loss.

**Key idea:**

- For **inter-class mixup**, standard Mixup applies: convex interpolation of both input and label.

- For **intra-class mixup**, only the dominant class label is amplified by a factor depending on $|0.5 - m|$ to encourage intra-class diversity while conserving class identity.

In this work, we set the intra-class amplification factor to $M = 0.2$. To prevent label probabilities from exceeding 1, we pre-process all labels by assigning the dominant class a value of $1 - M$, and the remaining $M$ is equally distributed across the other $C - 1$ classes.

---

**Algorithm 1** Generalized Mixup (G-Mixup) for a Mini-Batch

---

**Require:** Mini-batch $\{(x_i, y_i^{\text{raw}})\}_{i=1}^N$, mixup parameter $\alpha$, amplification factor $M$, number of classes $C$

1: **Preprocess labels:** for each $y_i^{\text{raw}}$, construct $y_i$ as:
   - $y_i[k] = 1 - M$, where $k = \arg\max(y_i^{\text{raw}})$
   - $y_i[\text{others}] = \frac{M}{C-1}$
2: Sample $m \sim \text{Beta}(\alpha, \alpha)$
3: Shuffle mini-batch to get $(x_i', y_i')$
4: Compute mixed inputs: $x_i^m = m x_i + (1 - m) x_i'$
5: **if** $\arg\max(y_i) == \arg\max(y_i')$ **(intra-class) then**
6:      $y_i^m[k] = y_i[k] + \frac{M}{2} - M \cdot |0.5 - m|$
7:      $y_i^m[\text{others}] = \frac{1 - y_i^m[k]}{C-1}$
8: **else**
9:      $y_i^m = m y_i + (1 - m) y_i'$ {Use $M$-adjusted labels}
10: **end if**
11: Compute prediction $\hat{y}_i = f(x_i^m)$
12: **if** loss is MSE **then**
13:      $\mathcal{L} = \frac{1}{N} \sum_i \|\hat{y}_i - y_i^m\|^2$
14: **else if** loss is NLL **then**
15:      $\mathcal{L} = \frac{1}{N} \sum_i \text{CrossEntropy}(\hat{y}_i, y_i^m)$
16: **end if**
17: Backpropagate $\mathcal{L}$

---

# G   Experimental Setup for Continual ImageNet

We evaluate *G-Mixup* on the Continual ImageNet benchmark, following a widely adopted binary task formulation. The full dataset used in this task consists of 1,000 classes, each with 700 images. For each class, 600 samples are used for training and 100 for testing. A continual learning sequence is constructed by randomly pairing classes to create binary classification tasks. Each task comprises 1,200 training samples and 200 test samples, drawn from two classes. Models are trained for 250 epochs using mini-batches of size 100. All images are downsampled to $32 \times 32$ resolution to reduce computational cost.

For all experiments, we adopt a convolutional neural network consisting of three convolutional layers with max-pooling, followed by three fully connected layers. To study the impact of representation dimensionality, we employ two variants: a wide network with larger hidden dimensions, identical to the one used in [7] (see Table 6), representing the high-dimensional regime; and a narrow network with reduced layer widths (see Table 5), corresponding to the low-dimensional case. The final output layer has two units representing the current task's classes. At the start of each task, the output heads are reset to zero—a common practice in continual learning benchmarks—although we note that this introduces privileged information about task boundaries. All methods are trained using SGD with momentum (set to 0.9), and models are initialized once at the beginning of the entire continual learning sequence. Learning rates are set as 0.01; results reported are averaged over 5 independent runs.

We compare *G-Mixup* with multiple baselines and LoP mitigation strategies. Two reference points are included: (1) **Reinit**, which re-initializes the model for every new task, and (2) **No Intervention (No Intv.)**, which applies naive continual training without any LoP countermeasures. For fair comparison,

We also include representative methods from three major strategies for mitigating LoP: **L2 Init** [17] (a regularization-based method, with the weight decay coefficient fixed at $5 \times 10^{-4}$), **LayerNorm** [21] (an architecture-based approach, where LayerNorm is applied before the activation function at every layer—including both convolutional and fully connected layers—except for the final output layer), and **CBP** [7] (a reset-based method).

Unless otherwise specified, all methods use the ELU activation function and MSE loss. The only exception is **CBP**, which uses ReLU as in its original implementation [7]. This choice is motivated by the fact that CBP relies on tracking the utility of individual neurons, and ELU—due to its non-zero minimum—can distort utility estimation by introducing persistent low-level activation. We further confirmed that CBP performs better with ReLU than with ELU under our experimental settings. All other hyperparameters and training schedules are kept consistent across methods to ensure fair comparison.

Table 5: `SmallConvNet` Architecture Details.

| Layer 1: Convolutional + Max-Pooling | | | |
|---|---|---|---|
| **Number of Filters** | 16 | **Activation** | ELU/Tanh/ReLU |
| **Convolutional Filter Shape** | (5,5) | **Convolutional Filter Stride** | (1,1) |
| **Max-Pooling Filter Shape** | (2,2) | **Max-Pooling Filter Stride** | (2,2) |
| Layer 2: Convolutional + Max-Pooling | | | |
| **Number of Filters** | 32 | **Activation** | ELU/Tanh/ReLU |
| **Convolutional Filter Shape** | (3,3) | **Convolutional Filter Stride** | (1,1) |
| **Max-Pooling Filter Shape** | (2,2) | **Max-Pooling Filter Stride** | (2,2) |
| Layer 3: Convolutional + Max-Pooling | | | |
| **Number of Filters** | 64 | **Activation** | ELU/Tanh/ReLU |
| **Convolutional Filter Shape** | (3,3) | **Convolutional Filter Stride** | (1,1) |
| **Max-Pooling Filter Shape** | (2,2) | **Max-Pooling Filter Stride** | (2,2) |
| Layer 4: Fully Connected | | | |
| **Output Size** | 16 | **Activation** | ELU/Tanh/ReLU |
| Layer 5: Fully Connected | | | |
| **Output Size** | 16 | **Activation** | ELU/Tanh/ReLU |
| Layer 6: Fully Connected | | | |
| **Output Size** | 2 | **Activation** | None (Linear) |

Table 6: `ConvNet` Architecture Details.

| Layer 1: Convolutional + Max-Pooling | | | |
|---|---|---|---|
| **Number of Filters** | 32 | **Activation** | ELU/Tanh/ReLU |
| **Conv. Filter Shape** | (5,5) | **Conv. Filter Stride** | (1,1) |
| **Max-Pool Filter Shape** | (2,2) | **Max-Pool Filter Stride** | (2,2) |
| Layer 2: Convolutional + Max-Pooling | | | |
| **Number of Filters** | 64 | **Activation** | ELU/Tanh/ReLU |
| **Conv. Filter Shape** | (3,3) | **Conv. Filter Stride** | (1,1) |
| **Max-Pool Filter Shape** | (2,2) | **Max-Pool Filter Stride** | (2,2) |
| Layer 3: Convolutional + Max-Pooling | | | |
| **Number of Filters** | 128 | **Activation** | ELU/Tanh/ReLU |
| **Conv. Filter Shape** | (3,3) | **Conv. Filter Stride** | (1,1) |
| **Max-Pool Filter Shape** | (2,2) | **Max-Pool Filter Stride** | (2,2) |
| Layer 4: Fully Connected | | | |
| **Output Size** | 128 | **Activation** | ELU/Tanh/ReLU |
| Layer 5: Fully Connected | | | |
| **Output Size** | 128 | **Activation** | ELU/Tanh/ReLU |
| Layer 6: Fully Connected | | | |
| **Output Size** | 2 | **Activation** | None (Linear) |

## H   Continual ImageNet Results

The performance of wide and narrow networks on Continual ImageNet is summarized in Table 7 and Table 8.

Table 7: SmallConv Test Accuracy Results on Continual ImageNet

| Task (×1000) | 0-1 | 1-2 | 2-3 | 3-4 | 4-5 |
|---|---|---|---|---|---|
| No Intv. | 0.817 (±0.070) | 0.805 (±0.085) | 0.562 (±0.089) | 0.500 (±0.000) | 0.500 (±0.000) |
| Retrained | 0.853 (±0.065) | 0.845 (±0.066) | 0.845 (±0.068) | 0.840 (±0.067) | 0.840 (±0.072) |
| L2 init | 0.804 (±0.055) | 0.796 (±0.059) | 0.786 (±0.054) | 0.785 (±0.052) | 0.788 (±0.055) |
| Layernorm | 0.753 (±0.056) | 0.760 (±0.052) | 0.759 (±0.051) | 0.751 (±0.049) | 0.751 (±0.056) |
| CBP | 0.834 (±0.052) | 0.847 (±0.052) | 0.846 (±0.050) | 0.847 (±0.050) | 0.857 (±0.052) |
| G-mixup | **0.866 (±0.052)** | **0.881 (±0.048)** | **0.885 (±0.048)** | **0.880 (±0.050)** | **0.879 (±0.054)** |

Table 8: ConvNet Test Accuracy Results on Continual ImageNet

| Task (×1000) | 0-1 | 1-2 | 2-3 | 3-4 | 4-5 |
|---|---|---|---|---|---|
| No Intv. | 0.794 (±0.063) | 0.778 (±0.073) | 0.604 (±0.135) | 0.537 (±0.074) | 0.500 (±0.000) |
| Retrained | 0.857 (±0.060) | 0.851 (±0.060) | 0.850 (±0.060) | 0.849 (±0.060) | 0.846 (±0.064) |
| L2 init | 0.814 (±0.047) | 0.805 (±0.050) | 0.800 (±0.052) | 0.803 (±0.052) | 0.807 (±0.054) |
| Layernorm | 0.782 (±0.052) | 0.768 (±0.056) | 0.752 (±0.049) | 0.749 (±0.050) | 0.755 (±0.055) |
| CBP | 0.848 (±0.053) | 0.867 (±0.049) | 0.864 (±0.046) | 0.863 (±0.048) | 0.878 (±0.047) |
| G-mixup | **0.875 (±0.049)** | **0.896 (±0.043)** | **0.899 (±0.042)** | **0.894 (±0.045)** | **0.896 (±0.047)** |

## I   FTLE Analysis of G-Mixup

We analyze the evolution of the FTLE distribution under Generalized Mixup at different stages of continual training. Our results reveal that G-Mixup consistently maintains the FTLE within a stable and moderate range throughout training. This controlled trajectory of representational dynamics effectively prevents both representation collapse and boundary chaotic behavior—two different types of LoP. The robustness of G-Mixup in sculpting the representation space is thus supported by its capacity to regulate FTLE evolution. The corresponding results are presented in Fig. 8.

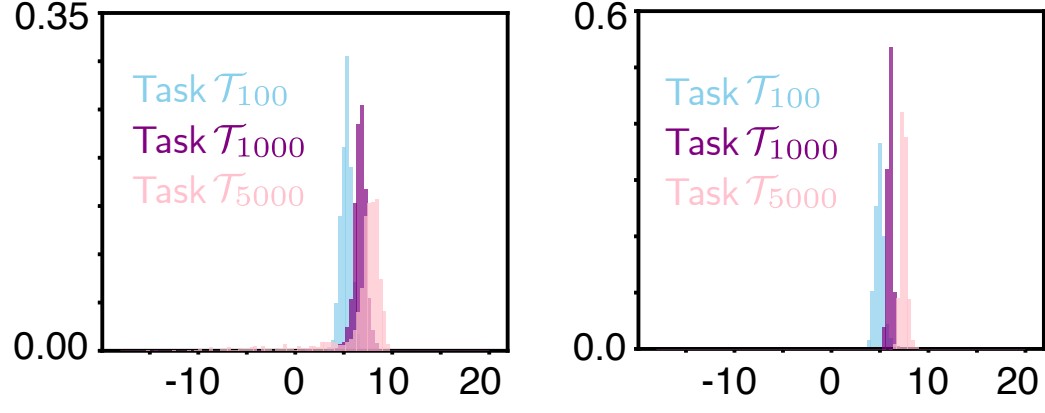

Figure 8: Analysis of FTLE distributions at different training stages of Generalized-mixup.

## J   Ablation Study

As discussed in the main text, Mixup encourages smoother transitions in the representation space by interpolating inter-class samples and labels, which helps mitigate Type-2 LoP (over-stretch). However, it leaves intra-class labels unchanged (fixed one-hot targets), limiting its ability to prevent

Type-1 LoP (collapse). In contrast, G-Mixup introduces controlled intra-class label variation , thereby regularizing both inter- and intra-class geometry.

To isolate and highlight this intra-class effect, we perform an ablation study in which both Mixup and G-Mixup are restricted to intra-class sample pairs only, removing inter-class interpolation entirely. This setup ensures that both methods generate the same level of sample diversity while differing only in their treatment of intra-class label variation. To accelerate the onset of collapse and better expose representational instability, we slightly increase the learning rate to 0.02.

As shown in Table 9, standard Mixup rapidly leads to collapsed representations, while G-Mixup continues to support progressive adaptation across tasks.

Table 9: Test Accuracy Results for Ablation Study

| Task ($\times 500$) | 0-1 | 1-2 | 2-3 | 3-4 |
|---|---|---|---|---|
| G-Mixup (SmallConv) | 0.864 | 0.871 | 0.879 | 0.878 |
| Mixup (SmallConv) | 0.864 | 0.581 | 0.500 | 0.500 |

## K   Class-Incremental CIFAR-100

We evaluate all methods under the class-incremental setting of CIFAR-100, where the model progressively learns 100 classes over 20 tasks, with 5 new classes introduced at each task. The model is trained on all accumulated classes at every stage and evaluated on the full set of seen classes, simulating task-agnostic continual learning. Each class contains 600 images, partitioned into 450 for training, 50 for validation, and 100 for testing.

For each increment, the model is trained for 200 epochs using SGD with a momentum of 0.9, weight decay of 0.0005, and a mini-batch size of 100. The learning rate is reset at the beginning of each increment and follows a decaying schedule: 0.1 for the first 60 epochs, 0.02 for the next 60, 0.004 for the following 40, and 0.0008 for the final 40 epochs. For G-Mixup, as the number of classes increases, inter-class interpolations become increasingly dense, making the representation space more entangled and the optimization landscape more complex. To stabilize training under this challenging regime, all learning rates are further reduced by a factor of 0.2 once the number of classes exceeds 10. The same learning-rate schedule is applied to all other baseline methods for fairness. However, under this schedule, their performance deteriorates markedly. Therefore, we present the results obtained under their best-performing configurations. During training, the best-performing model on the validation set is saved at each increment, and training for the next increment starts from this checkpoint, effectively applying early stopping across increments.

We adopt ResNet-18 as the backbone architecture and evaluate two variants: a standard high-dimensional version (ResNet-18) and a compressed low-dimensional variant (0.25× ResNet-18), where all channel widths and the final fully connected layer are reduced proportionally. Additionally, a fully connected layer with a width of 16 is appended as the representation layer in the low-dimensional variant. The output layer grows dynamically by adding 5 new units at each increment, with newly added weights initialized via Kaiming initialization and biases set to zero. Convolutional and linear layers follow Kaiming initialization, while batch normalization weights are set to 1.

All input images are normalized to [0, 1], channel-wise standardized using dataset statistics, and augmented during training with random horizontal flips, 4-pixel padding followed by random crops, and random rotations between $[-15°, 15°]$. These augmentations are applied only to the training set.

We compare *G-Mixup* against multiple baselines and LoP mitigation strategies under the class-incremental CIFAR-100 setting. Two primary baselines are considered: (1) **No Intervention (No Intv.)**, which performs standard continual training without any mechanisms to counteract LoP, and (2) **Reinit**, which re-initializes the model from scratch at each task.

To ensure a comprehensive evaluation, we further include representative methods from three major categories of LoP mitigation: **L2 Init** [17], a regularization-based method with a fixed weight decay of $5 \times 10^{-4}$ and no additional L2 regularization on the weight; **LayerNorm** [21], an architecture-based approach where BatchNorm is replaced with LayerNorm in all layers; and **CBP** [7], a reset-based approach targeting underutilized neurons. For fair comparison, and to rule out the possibility that

G-Mixup's improvement originates from label softening rather than its geometric regularization effect, all baseline methods are trained with the same soft-label smoothing coefficient of 0.1.

All methods are evaluated under identical training protocols as described above. The comparative results are presented below.

Table 10: 0.25x Resnet-18 Acc. on CIFAR100

| Task (×4) | 0-1 | 1-2 | 2-3 | 3-4 | 4-5 |
|---|---|---|---|---|---|
| No Intv. | 0.927 (±0.027) | 0.812 (±0.018) | 0.754 (±0.015) | 0.708 (±0.014) | 0.661 (±0.008) |
| Retrained | 0.926 (±0.035) | 0.800 (±0.020) | 0.745 (±0.013) | 0.702 (±0.012) | 0.666 (±0.007) |
| L2 init | 0.925 (±0.032) | 0.788 (±0.020) | 0.724 (±0.013) | 0.682 (±0.014) | 0.649 (±0.007) |
| Layernorm | 0.851 (±0.051) | 0.755 (±0.028) | 0.723 (±0.016) | 0.674 (±0.015) | 0.641 (±0.008) |
| CBP | 0.923 (±0.038) | 0.812 (±0.020) | 0.760 (±0.017) | 0.713 (±0.011) | 0.678 (±0.006) |
| G-mixup[ours] | **0.932 (±0.030)** | **0.825 (±0.019)** | **0.772 (±0.014)** | **0.732 (±0.014)** | **0.697 (±0.007)** |

Table 11: Resnet-18 Acc. on CIFAR100

| Task (×4) | 0-1 | 1-2 | 2-3 | 3-4 | 4-5 |
|---|---|---|---|---|---|
| No Intv. | 0.907 (±0.029) | 0.847 (±0.014) | 0.812 (±0.013) | 0.778 (±0.013) | 0.743 (±0.008) |
| Retrained | 0.901 (±0.037) | 0.842 (±0.017) | 0.815 (±0.012) | 0.788 (±0.013) | 0.767 (±0.005) |
| L2 init | 0.922 (±0.027) | 0.829 (±0.016) | 0.785 (±0.013) | 0.752 (±0.012) | 0.723 (±0.006) |
| Layernorm | 0.847 (±0.065) | 0.782 (±0.035) | 0.763 (±0.019) | 0.728 (±0.019) | 0.697 (±0.019) |
| CBP | 0.907 (±0.027) | 0.847 (±0.015) | 0.816 (±0.013) | 0.788 (±0.011) | **0.768 (±0.008)** |
| G-mixup[ours] | **0.928 (±0.025)** | **0.864 (±0.016)** | **0.832 (±0.012)** | **0.800 (±0.009)** | **0.768 (±0.006)** |

## L   Computing Infrastructure

Table 12: Computing infrastructure of the primary server

| | |
|---|---|
| **CPU** | AMD EPYC 9654 96-Core Processor, 2 sockets (384 threads total) |
| **GPU** | NVIDIA RTX 4090 |
| **Memory** | 512 GB |
| **Operating system** | Ubuntu 20.04.6 LTS |
| **Simulation platform** | Python 3.11 with PyTorch 2.1.1 |

Table 13: Computing infrastructure of the secondary server

| | |
|---|---|
| **CPU** | AMD EPYC 9354 |
| **GPU** | NVIDIA RTX 4090 |
| **Memory** | 512 GB |
| **Operating system** | Ubuntu 20.04.6 LTS |
| **Simulation platform** | Python 3.11 with PyTorch 2.1.1 |

The simulations and analyses in this study are conducted on two high-performance computing servers equipped with AMD EPYC processors and NVIDIA RTX 4090 GPUs. The primary server features dual AMD EPYC 9654 CPUs (each with 96 cores), totaling 384 threads, and supports simultaneous multithreading. The secondary server is equipped with an AMD EPYC 9354 CPU and identical GPU and software configuration. Both systems are configured with 512 GB of memory, enabling efficient execution of memory-intensive simulations and deep learning workloads. Experiments were run under Ubuntu 20.04.6 LTS using Python 3.11 and PyTorch 2.1.1, providing a robust and up-to-date scientific computing environment.

