# OpenReview forum: "The Dual Nature of Plasticity Loss in Deep Continual Learning: Dissection and Mitigation"
_NeurIPS.cc/2025/Conference — NeurIPS 2025 poster_

### Official Review · Reviewer_GUee · 2025-06-27

**Clarity:** 4
**Significance:** 3
**Originality:** 3
**Rating:** 5
**Confidence:** 4

**Summary:**

This paper presents a novel framework for understanding and mitigating the Loss of Plasticity (LoP) in deep continual learning.  The authors posit that LoP is a dual-natured problem, dissecting it into two distinct types. Specifically, Type-1 LoP is characterized by the collapse of the representation space within a class; Type-2 LoP is characterized by an over-stretching of the representation space near decision boundaries.  Based on this new understanding, they propose Generalized Mixup (G-Mixup), an extension of the standard Mixup technique. G-Mixup is designed to specifically counteract both types of LoP by not only regularizing the space between classes but also actively expanding the representation space within a class to prevent collapse. The authors validate their theory and the effectiveness of G-Mixup through experiments on real-world datasets, including ImageNet and CIFAR-100, where G-Mixup is shown to outperform existing LoP mitigation techniques.

**Questions:**

1. It is well-known that certain regularization-based methods designed to mitigate forgetting (e.g., EWC[3], LwF[4]) can inadvertently harm plasticity. It is not immediately clear how this phenomenon fits into the proposed dual-nature framework. Could you elaborate on whether the plasticity reduction caused by such stability-focused methods can be explained as a form of Type-1 (collapse) or Type-2 (chaos) LoP, or if it represents a different mechanism altogether?

[3] Overcoming catastrophic forgetting in neural networks. PNAS 2017

[4] Learning without forgetting. TPAMI 2017

**Ethical Concerns:**

["NO or VERY MINOR ethics concerns only"]

**Final Justification:**

Given that this paper provides an insightful analysis of the loss of plasticity and proposes a practical solution, I believe it makes a valuable contribution to the field of continual learning and deserves acceptance.

**Limitations:**

yes

**Quality:**

4

**Strengths And Weaknesses:**

**Strengths**
1. The paper's key contribution is its insightful reframing of LoP . By categorizing LoP into two distinct types based on the dynamics of representation space (collapse vs. chaotic expansion), the work offers a convincing and novel perspective on this challenging problem. The empirical evidence presented, particularly in Figure 4, provides strong support for this dual-nature hypothesis.
2. The paper is exceptionally well-written and easy to follow. The authors effectively use clear visualizations (e.g., Figures 2 and 3) to build intuition around the core theoretical concepts
3. The proposed G-Mixup method is directly and logically derived from the theoretical analysis. The experimental results confirm its effectiveness across different settings.

**Weaknesses**
1. A direct comparison between G-Mixup and Mixup is crucial for validating the central claims of the paper. I suggest the authors add this comparison to the **main text** of this paper.
2. The paper focuses almost exclusively on plasticity, which is its stated goal. However, in continual learning, plasticity is inherently linked to stability (i.e., mitigating catastrophic forgetting). A more comprehensive evaluation would benefit from a discussion of how G-Mixup affects previously learned tasks. Evaluation metrics could refer to [1] [2].

[1] Rethinking the Stability-Plasticity Trade-off in Continual Learning from an Architectural Perspective. ICML 2025

[2] Improving plasticity in online continual learning via collaborative learning. CVPR 2024

---

> ### Author Rebuttal · Authors · 2025-07-31
>
> We sincerely thank the reviewer for their time, thoughtful feedback, and positive assessment of our work. We are particularly grateful for the encouraging remarks regarding the novelty and significance of our contributions. We address the questions in detail below and are happy to further discuss any remaining concerns.
>
> **Weaknesses 1**: We are grateful to the reviewer's suggestion on adding the comparison between Mixup and G-Mixup to the main text, which is indeed crucial for demonstrating our method's true effectiveness. As pointed out in the main text mixup promotes smooth transitions in the repr. space by generating interpolated inter-class samples and labels, which can mitigate Type-2 LoP (i.e., over-stretch). However, it retains unchanged intra-class labels (i.e., fixed one-hot targets), which limits its ability to prevent Type-1 LoP. In contrast, G-mixup introduce intra-class label difference through Eq. 15.
>
> Therefore, to clearly demonstrate the difference between the effect of the two mixup-style methods. We conduct an additional experiment that restrict mixup and G-mixup to intra-class sample pairs only, removing the inter-class smoothing effects for both party, and focus on the intra-class repr. space where mixup and G-Mixup provide the same amount of sample diversity but G-Mixup further promotes intra-class representation enhancement.
>
> | Task(x500) | 0-1    | 1-2    | 2-3    | 3-4    |
> | ---------- | ------ | ------ | ------ | ------ |
> | G-Mixup    | 0.8641 | 0.8715 | 0.8792 | 0.8778 |
> | Mixup      | 0.8635 | 0.5813 | 0.5000 | 0.5000 |
>
> As shown in the table above, mixup rapidly leads to collapsed representations and stagnated learning, whereas G-Mixup continues to support learning subsequent tasks. This result clearly demonstrates the critical role of G-Mixup in addressing LoP, apart from providing sample diversity. We will include these results in the main text of the revised manuscript.
>
> **Weaknesses 2**: We thank the reviewer for raising this insightful point. As correctly noted, the stability–plasticity dilemma is fundamental in continual learning. Our study explicitly focuses on _plasticity loss_, a relatively new but increasingly important perspective that emphasizes the degradation of a network’s capacity to adapt to new tasks, even in the absence of explicit constraints to retain prior knowledge.
>
> Benchmarks such as Continual ImageNet, which involve thousands of sequential tasks (up to 5,000), push the network into regimes where backward consistency is often infeasible or undesirable. In such settings, many methods—including CBP—explicitly reset parts of the model after each task, sacrificing stability in favor of forward plasticity. Similar motivations are found in reinforcement learning, where preserving previously learned value functions from outdated policies may hinder rather than help future performance. Therefore, in our work, the primary objective is to ensure that the network retains sufficient plasticity to continue adapting to novel tasks, rather than preserving accuracy on past tasks.
>
> That said, we fully agree that empirical assessments of stability are important. In response to the reviewer's suggestion, we report two key metrics of stability—Average Forgetting (AF) and Relative Forgetting (RF)—as defined in the reviewer's referenced work [1], to better illustrate how performance on previous tasks may deteriorate. Furthermore, since Continual ImageNet is formulated as a binary classification problem, we additionally report the average deviation from chance level (i.e., 0.5) after learning all 5000 tasks, reflecting the network’s retained knowledge on previously encountered tasks.
>
> Table A Stability metrics on Continual ImageNet using SmallConvNet
> | Algo            | AF   | RF   | Std |
> | --------------- | ---- | ---- | ---------------- |
> | L2 init         | 0.25 | 0.32 | 0.15      |
> | CBP             | 0.23 | 0.29 | 0.12      |
> | G-mixup [Ours] | 0.30 | 0.36 | 0.13      |
>
> Table B Stability metrics on Continual ImageNet using ConvNet
> | Algo            | AF   | RF   | Std |
> | --------------- | ---- | ---- | ------ |
> | L2 init         | 0.27 | 0.33 | 0.15      |
> | CBP             | 0.27 | 0.33 | 0.07      |
> | G-mixup [Ours] | 0.28 | 0.34 | 0.11      |
>
> As shown in the tables above, all methods experience some degree of forgetting on previously learned tasks. When evaluating the final model—after training on all 5000 tasks—on earlier tasks, we observe different behaviors in terms of variance from the 0.5 chance-level baseline:
>
> - **CBP** exhibits the lowest variance, suggesting that its representations are largely decorrelated from earlier task classifiers. This is consistent with the fact that CBP resets part of the network during each task, leading to representations that are less aligned with previously learned classification heads.
> - **L2 init** achieves the highest variance. This reflects the fact that L2 init constrains weights to stay close to their initialization, which helps maintain representational stability, but its performance largely depends on how well the representations induced by the initial weights align with the current task.
> - **G-Mixup** shows higher AF and RF values, as it allows the representation space to adapt more freely in order to optimize performance on the current task. While this may introduce some deviation from earlier representations, it supports effective continual learning by maintaining sufficient plasticity.
>
>
> **Question**: Regarding the reviewer’s thought-provoking question about the relationship between regularization-based stability methods (such as EWC and LwF) and our dual-nature framework of plasticity loss:
>
> While these methods also lead to impaired plasticity, the underlying mechanisms are fundamentally different. Regularization-based techniques impose constraints on network dynamics—e.g., freezing important weights (EWC) or forcing new predictions to remain close to prior outputs (LwF)—which restrict the network's ability to adapt. In contrast, _loss of plasticity_ as defined in our work refers to a spontaneous decline in adaptability that emerges _even without any external constraints_, purely due to cumulative learning and optimization dynamics.
>
> Therefore, the plasticity reduction caused by stability-focused methods does not neatly fall into the Type-1 (representational collapse) or Type-2 (representational chaos) categories we describe. Instead, they represent a separate, externally induced mechanism that trades plasticity for stability by design.
>
> Nevertheless, there is an interesting conceptual overlap. Some recent studies in the plasticity loss literature have borrowed ideas from stability methods to _preserve plasticity_. For example:
>
> - **L2Init**: a plasticity-preserving counterpart to EWC that penalizes deviations from initial weights, under the hypothesis that the initial network state is maximally plastic.
> - **Initial Feature Regularization** (e.g., Lyle et al. 2022): constraining representations to remain close to those from random networks (conceptually similar to LwF), preserving expressivity and adaptability.
>
> We sincerely thank the reviewer for these important references and suggestions. We will include this discussion in the revised manuscript to clarify the relation between our framework and stability-based methods, and to enhance the theoretical richness of our work.

---

> > ### Comment · Reviewer_GUee · 2025-08-05
> >
> > I thank the authors for their comprehensive response, which has effectively addressed my primary concerns. Given that this paper provides an insightful analysis of the *loss of plasticity* and proposes a practical solution, I believe it makes a valuable contribution to the field of continual learning and deserves acceptance.

---

> > > ### Author Response · Authors · 2025-08-05
> > >
> > > We sincerely thank the reviewer for the constructive advice and for taking the energy and time in active discussion. We greatly appreciate recommendation. Your insightful comments were instrumental in guiding significant improvements and refinements to the paper.

---

### Official Review · Reviewer_m8rC · 2025-06-30

**Clarity:** 3
**Significance:** 2
**Originality:** 2
**Rating:** 4
**Confidence:** 4

**Summary:**

The paper utilizes the unconstrained feature model (UFM) and finite-time Lyapunov exponent (FTLE) to investigate loss of plasticity. Two different types have been found: 1) decreasing FTLE, 2) increasing FTLE. Both types can coexist and are affected by the dimensionality of the network. Based on these theories, the paper proposed G-mixup as a way to smooth the between-class differences and increase the within-class differences. Such a method outperforms the existing methods.

**Questions:**

N/A

**Ethical Concerns:**

["NO or VERY MINOR ethics concerns only"]

**Final Justification:**

The authors present a compelling, geometry-inspired analysis of “loss of plasticity” in continual learning—using finite-time Lyapunov exponents (FTLE) to distinguish between representation collapse and chaos—and couple it with a simple yet effective Generalized Mixup strategy. In the original version, the connection between the strategy and the theory is not strongly established. However, in the authors' rebuttal, they provide a direct method for mitigating LoP through controlling FTLE. There remains a theoretical gap between FTLE and accuracy, but the authors offer an intuitive explanation in the rebuttal, and hopefully future work can address it. As pointed out by Reviewer FyTC, catastrophic forgetting is heavily involved in continual learning, while LoP is less common. This reduces the overall significance of the current paper. In sum, I rate it as borderline acceptable.

**Limitations:**

yes

**Quality:**

4

**Strengths And Weaknesses:**

Strengths:
1) The paper proves that the training will causes a compression/stretching behavior in the penultimate layer representations.
2) The paper finds a marker for LoP (a large FTLE causes chaotic behavior, and a small FTLE causes compression).
3) The paper proposes a new method (G-mixup) to mitigate LoP by smoothing the boundaries and increasing the variability within the class.

Weakness:
1) FTLE in the framework seems unimportant: the major theorem 3.1.2 (since both type 1 and type 2 lop are defined on it) only introduces FTLE after the whole proof to reduce the notation of stretching/compressing $h$ representations.
2) The paper gives much proof over the stretching/compressing behavior of the penultimate layer representations, but there is a gap between the behavior of $h$ and LoP. It would strengthen the paper if one could prove that the accuracy will decrease if the sample follows in the stretching/compressing region, and will cause the FTLE to increase or decrease. It would also strengthen the paper if one could empirically show that when the sample follows in the stretching/compressing region, and will cause the FTLE to increase or decrease (i.e., L227 and L228). The current paper shows an overall relationship between FTLE and LoP, but this does not connect to the theory.
3) Empirically, the authors show G-mixup can mitigate the LoP and control the range of FTLE. To more closely relate to the theorem, it would be better to show results with directly controlling FTLE (e.g., set FTLE as a penalty if it deviates too much from task 1, or dropping some samples if it causes a too wild FTLE).

---

> ### Author Rebuttal · Authors · 2025-07-31
>
> We sincerely thank the reviewer for their thorough reading, thoughtful feedback, and constructive suggestions. We deeply appreciate the time and effort the reviewer devoted to evaluating our work. We have carefully addressed the reviewer’s comments in the following rebuttal, and we hope our clarifications successfully resolve the raised concerns.
>
> **Weakness 1**
> In our work, FTLE is introduced only to quantify rather than to understand how the geometry of representation evolves during continual learning. Therefore, we'd like to argue that this is not a weakness of our work since our main contributions and emphases have been on the understanding of how the geometry of representational space evolves during continual learning, rather than FTLE itself.
>
> That said, we still want to justify the application of FTLE in our work. FTLE is a widely adopted metric used in physics for describing the spatio-temporal dynamics of various systems, and was first introduced to deep learning analogously by Storm et. al., 2024 (ref 27) to describe how distance in the representational space relates to the distance in the input space. Here, we further develop a theoretical framework utilizing this metric, specifically for the dynamical characterization of the local geometry of representational space during continual learning. It has successfully described how the distance between representations of same-class samples and different-class samples evolves as learning continues. Therefore, we think it is fair to say that FTLE is very essential to our work as a quantifying tool.
>
> **Weakness 2**
> We thank the reviewer for this insightful suggestion, which indeed helps strengthen the connection between the theoretical framework and empirical observations in our study.
>
> We address this concern from both an _intuitive_ and an _empirical_ perspective:
>
> #### 1. Theoretical Intuition
>
> The penultimate layer representation $h(x)$ is fed into a linear classifier to perform final prediction. For the linear classifier to perform well, representations must be linearly separable. Therefore:
>
> - In **compressing regions** of representational space with lower effective dimensions (prone to Type 1 LoP), if representations from _different classes_ are mapped to nearby or overlapping points (i.e., the compression collapses multiple classes into the same representation space), linear classification becomes impossible, leading to _accuracy degradation_.
>
> - In **stretching regions** of representational space of higher effective dimensions (prone to Type 2 LoP), small input perturbations are mapped to significantly different representations. When _same-class samples_ are send apart in various directions, the local geometry become distorted, making it difficult for the classifier to assign consistent predictions, again leading to lower _accuracy_.
>
> These effects suggest that regions with extreme FTLE values (on either end) correspond to representational spaces where the local geometry make same-class alignment and inter-class separation progressively infeasible, both are detrimental to classification performance.<!--  Due to time and character limit, we are unable to provide a detailed proof, however we believe a rigorous rewrite of these intuitive understandings into mathematical proofs is possible. -->
>
> #### 2. Empirical Verification
>
> To further substantiate this link, we visualize classification accuracy over training time for samples grouped by FTLE ranges. Initially, when LoP has not emerged, FTLE values are small and classification accuracy is high. As LoP progresses, FTLE values become either too large or too small (strong stretching/compression), and classification accuracy for these regions drops significantly. These findings empirically support the theoretical link between FTLE, local representational distortion, and classification accuracy.
>
> ####  Accuracy and Sample Distribution within Different FTLE Ranges across Tasks
>
> | Task id |FTLE range      | <-10       | -10~0      | 0~10        | >10         |
> |-------|-------------|------------|------------|-------------|-------------|
> | 100   | Accuracy    | 0.000      | 0.000      | 0.800       | 0.000       |
> |       | Sample Count| 0          | 0          | 200         | 0           |
> | 1000  | Accuracy    | 0.000      | 0.000      | 0.813       | 0.804       |
> |       | Sample Count| 0          | 0          | 16          | 184         |
> | 5000  | Accuracy    | 0.518      | 0.000      | 0.000       | 0.713       |
> |       | Sample Count| 112        | 0          | 1           | 87          |
>
> We thank the reviewer again for pointing out this gap, and we will include these new results in the appendix to clarify this connection between LoP dynamics and performance impact.
>
> As for the other empirical support suggestion: "if one could empirically show that when the sample follows in the stretching/compressing region, and will cause the FTLE to increase or decrease", we believe there's a misunderstanding. As proved by theorem 3.1.2, same-class samples decrease the FTLE values for the region of interest while different-class samples increase them. This happens regardless of whether the samples are following the stretched or compressed region (through the force of back-propagation). Then based on which pair of samples (same-class or different-class) are put into these regions, the samples become less likely to be correctly classified, as supported by the theoretical intuition and empirical support above.
>
> **Weakness 3**
> We thank the reviewer for this highly insightful suggestion. Indeed, when initially designing our approach, we also considered directly penalizing the FTLE in the loss function to preserve plasticity. However, we ultimately opted for G-Mixup due to practical feasibility.
>
> Directly computing FTLE involves evaluating the local Jacobian of the feature extractor and estimating its maximum singular value, which is computationally expensive and may introduce gradient stability issues during training. This may make it hard and costly to use FTLE as a regularizer in large-scale settings like ImageNet.
>
> That said, inspired by the reviewer’s suggestion, we implemented a **proof-of-concept FTLE regularization experiment** in a toy model. Specifically, we approximated the FTLE for each sample using the following procedure:
>
> Let $h(x)$ denote the feature representation of input $x$, and let $J = \nabla_x h(x)$ be the local Jacobian. We approximate the FTLE using power iteration:
>
> $$
> \\begin{aligned}
> &\\text{Initialize } v \\sim \\mathcal{N}(0, I), \\quad v \\leftarrow \\frac{v}{\\|v\\|} \\
> &\\text{Iteratively update:} \\quad v \\leftarrow \\frac{J^\\top (Jv)}{\\|J^\\top (Jv)\\|} \\
> &\\text{Then:} \\quad \\text{FTLE}(x) = \\log \\left( \\| Jv \\| + \\varepsilon \\right)
> \\end{aligned}
> $$
>
> We then add the following regularization term to the loss:
>
> $$
> L_{\\text{FTLE}} = \\lambda_{\\text{FTLE}} \\cdot \\left( \\text{FTLE}(x) \\right)^2
> $$
>
> This term penalizes large deviations of the FTLE from 0, encouraging local smoothness in representation dynamics.
> In practice, to reduce computational cost, we used a 20-layer network in the toy experiment, which tends to collapse easily under the setup of Toy Experiment 1 as shown in Figure 2 of the main text. In contrast, when FTLE regularization is applied, the network is able to continually adapt to new tasks without collapsing. To make the FTLE computation more efficient, we performed 5 iterations per sample during the power iteration procedure. Additionally, we employed a linearly increasing FTLE regularization coefficient, set as \\( \\lambda_{\\text{FTLE}} = 0.05 \\times \\text{epoch} \\) (per task), to balance task learning and the structure of the representational space.
>
> As shown in the table below, the FTLE regularization effectively mitigates LoP in toy settings. We thank the reviewer for the insightful suggestion and will discuss this alternative approach in the revised main text.
>
> #### Classification Accuracy over Task Ranges
>
> | Task Range | 1–10  | 11–20 | 21–30 | 31–40 | 41–50   |
> |---|-------|----|-----|-------|---------|
> | $ L_{classification} $        | 0.829 | 0.761 | 0.603 | 0.584 | 0.565   |
> | $ L_{classification} + L_{FTLE}$| 0.920 | 0.969 | 0.942 | 0.968 | 0.971 |
>
>
>
> We thank the reviewer for this valuable suggestion. We will include these results in the appendix and highlight the discussion in the final version. Exploring scalable and efficient FTLE-regularization is an exciting direction for future work.

---

> > ### Comment · Reviewer_m8rC · 2025-08-05
> >
> > I appreciate the authors’ thorough reply, which has fully resolved my main concerns.  I decided to keep my original score.

---

> > > ### Author Response · Authors · 2025-08-06
> > >
> > > We are glad to hear that our rebuttal has fully resolved the reviewer's concern, and sincerely thank the reviewer for the constructive advices and insightful comments, which have been instrumental in improving the clarity and the rigour of our work.

---

### Official Review · Reviewer_FyTC · 2025-07-01

**Clarity:** 2
**Significance:** 2
**Originality:** 3
**Rating:** 4
**Confidence:** 4

**Summary:**

The authors introduce an analytical perspective on why deep networks lose their ability to learn new tasks during continual training by combining Finite-Time Lyapunov Exponent (FTLE) analysis with the Unconstrained Feature Model from Neural Collapse theory. They show that loss of plasticity” (LoP) arises in two distinct modes, Type 1, where representations collapse within each class (highly negative FTLEs), and Type 2, where feature maps near decision boundaries become chaotic (highly positive FTLEs). They verify these phenomena on synthetic XOR/hyper-plane tasks and on Continual ImageNet and class-incremental CIFAR-100. Building on this insight, the authors propose Generalized Mixup, a simple extension of Mixup that amplifies labels during same-class interpolation to prevent collapse while still regularizing boundary regions, thereby stabilizing FTLEs.

**Questions:**

Q1. Why do you employ soft (i.e. non-one-hot) original labels in the Mix-up? What benefits does this confer?

Q2. Did you apply the same ‘preprocess labels’ stage to all baseline methods? Please clarify.

Q3. How can the activation layer operation be represented as a matrix multiplication? Please provide a clear justification or derivation.

Q4. Please clarify the definition of $\lambda_{trained,T}^{k,k}$ in section 3.2, especially the validity of $\lambda_0^{k,k}$.

Q5. Could you show the FTLE curves for the other baseline techniques as well?

Q6. Your theoretical analysis and proofs appear limited to binary-class settings. Can your key theorems and FTLE-based findings be generalized to multi-class classification, and if so, can you provide either theoretical extensions or empirical support?

Q7. Please report backward-transfer or average accuracy by periodically re-evaluating earlier tasks (or using a shared, growing head) to show whether your method also preserves past knowledge, not just forward plasticity

Q8. How sensitive is your method to the confidence-amplification factor M and to the choice of Beta priors for inter- and intra-class mixing?

**Ethical Concerns:**

["NO or VERY MINOR ethics concerns only"]

**Final Justification:**

Although the issue of catastrophic forgetting remains to some extent, the authors have provided satisfactory responses. Therefore, I maintain my positive score.

**Limitations:**

yes

**Paper Formatting Concerns:**

No formatting concerns.

**Quality:**

2

**Strengths And Weaknesses:**

Strengths:

S1. The authors quantify how the representation space transforms during continual training with theoretical analysis as well as experimental results.

S2. The paper provides an intuitive explanation of Type-1 and Type-2 mechanisms, along with a figure that makes them immediately understandable. Moreover, the concepts of FTLE and neural collapse (NC) are aptly extended to the continual learning setting

Weaknesses:

W1. In standard classification tasks, labels are represented as one-hot vectors. However, in the proposed Mix-up scheme, the original labels are not one-hot, and the manuscript does not explain why this choice is made.

W2. It is unclear whether the ‘preprocess labels’ stage was applied to the baselines in the experiments. Without this clarification, the experimental validity is questionable—if the baselines also perform the same preprocessing, their degraded performance might be an artifact of that step.

W3. Activation layers are typically expressed as element-wise nonlinear functions applied to features, because they cannot be written as matrix multiplications. Yet Equation (4) models the activation as a matrix product; this needs to be corrected or justified. Moreover, the framework does not explicitly account for front-end non-linearities (e.g., BatchNorm, residual scaling, optimizer dynamics), and the approximation errors resulting from these simplifications are not clearly quantified or empirically examined.

W4. It is not specified whether the continual learning scenario involves binary classification with labels that merely flip at each task step, or whether entirely new labels (i.e.\ new classes) are introduced. If the latter is the case, then in Section 3.2’s definition of $\lambda_{trained,T}^{k,k}$, the term $\lambda_0^{k,k}$ cannot exist: the first k refers to the initial label and the second to the label at task T, and they cannot be the same.

W5. The paper presents the FTLE curves for only the proposed method. Without showing the FTLE for the other baselines, the credibility of the claim is weakened.

W6. The analysis focuses on representation geometry, yet other LoP drivers—e.g., weight over-consolidation, dead-unit accumulation, curvature stiffening, learning-rate decay, or sheer capacity shortage—are neither modelled nor monitored. As a result, a system might lose plasticity even when FTLE tails look healthy.

W7. The evaluation is restricted to image classification, so it is unclear whether the reported gains are due to mitigating Type-1/2 LoP specifically or merely reflect the general regularization benefits of Mixup-style augmentations.

W8. By resetting classifier heads after each task, the study does not measure performance on prior tasks, leaving the risk of unobserved catastrophic forgetting.

---

> ### Author Rebuttal · Authors · 2025-07-31
>
> First of all, we'd like to thank the reviewer for the invaluable suggestions, especially the ones regarding soft labels, LoP mitigation vs. mixup-style augmentation and Forgetting Measure, together they help improve the quality and impact of our work. We will make sure to include the results here into the revised manuscript.
>
> **W1, W2 & Q1, Q2**
> In our original experiments (expt's), we did not preprocess the baseline methods with soft labels. To address this concern, we have now conducted additional expt's on both ImageNet and CIFAR-100 datasets, where all baselines are also trained using soft labels. On ImageNet with MSE loss, the impact of soft labels is negligible. On CIFAR-100, where cross-entropy(CE) loss is applied, soft labels offer a modest improvement during the learning of later tasks. This is because that soft labels help prevent the representational (repr.) collapse or over-expansion that can occur when logits are trained to match one-hot labels. Due to character limitations, we present results using CE loss as a representative case in this section. However, we have conducted all the experiments on the benchmarks presented in the main text, and would be happy to share those results upon request.
> For all methods, the intra-class soft label smoothing factor was consistently set to 0.1 throughout our experiments. With these additional expt's, we can now show that G-Mixup consistently outperforms all baselines regardless of whether soft labels are used, as shown in the following table:
>
> **Resnet-18 Acc. on CIFAR100 with SoftLabel**
> |Task(×10)|0-1|1-2|
> |--|--|--|
> |NoIntv.|0.850|0.813|
> |Retrained|0.856|0.817|
> |L2init|0.860|0.783|
> |Layernorm|0.848|0.773|
> |CBP|0.836|0.821|
> | **G-mixup[ours]** | **0.884** | **0.828** |
>
> Regarding the use of soft labels in G-Mixup (assuming the reviewer refers to our method): G-Mixup was originally designed for MSE loss, using one-hot interpolation for inter-class mixing and
> (1 + $\alpha$,0)–style labels for intra-class mixing. To enable the application of G-Mixup under the CE loss, we adopt soft labels to ensure the label remains a valid probability distribution (i.e., sums to one). This allows same-class samples to have different targets based on the mixing ratio m (see Eq. 15).
>
> **W3 & Q3**
>
> Indeed, real-world DNN contain layers with nonlinear activation where matrix multiplications cannot be directly applied. To circumvent this complication, we adopt unconstrained feature model (UFM, Eq. 4), where the post-activation representations are treated as free, trainable variables. This widely adopted simplification have been proven effective and necessary to advance the theoretical study of DL, especially for works on neural collapse and repr. dynamics (Zhu et. al., 2021 NeurIPS (ref. 34)). It allows us to reduce DL to an optimization problem between the last two layers, enabling analytical investigation of repr. dynamics. Finally, we have justified this with numerical experiments of both toy models and real-world datasets.
>
> As the reviewer points out, nonlinearities—including BatchNorm, residual scaling, and the choice of optimizer—can all affect the learned representation. While our theoretical framework does not explicitly model these, we did conduct empirical validations under these settings, suggesting the simplified approach still works in the real-world setting. Specifically:
>
> - CIFAR-100 benchmark uses both BatchNorm and ResNet, thereby incorporating BN and skip connections.
> - On ImageNet, we test our method under various optimizer, including SGD, Adam, and AdamW.
>
> **W4 & Q5**
>
> In our setting, we consider a continual learning (CL) scenario where for each new task $T$, the two units in the final-layer are reset to zeros. It could be said that these are new labels (the later case), but actually they are still $[1, 0]$ or $[0, 1]$ in the end (the former case). However, we want to emphasize that the notation $k$ does not concern with the labels, rather, they refer to the input distributions associated with the class. Note that in our CL setups, we allow the samples from a class to reappear and participate in multiple tasks. These scenarios occurs frequently in real-world CL when the definition of a single concept is learnt gradually against other concepts to form the map of knowledge, it is also frequently encountered in RL when policy rules changes under different context while the object class remains the same.
>
> Therefore, what's important is that any given pair of input samples of the same class $k$ share the same label in each task regardless of the vector being $[1, 0]$ or $[0, 1]$. This ensures their representations are consistently driven toward alignment and away from samples of other classes.
>
> In conclusion, the use of $\lambda_0^{k,k}$ and $\lambda_{trained,T}^{k,k}$ is well justified regardless of label encoding. The differences in labels across tasks primarily affect the classifier weights, but not the underlying repr. structure in the feature space.
>
> **W5, W6 & Q6**
> Presenting FTLE curves for other methods would indeed strengthen our claims. However, due to the NeurIPS rebuttal format restrictions and character limitations, we only report the FTLE distribution statistics for baseline methods in the table below. Results for other methods are also available and can be provided upon request.
>
> Table: FTLE Distribution Across Tasks for Different Methods (ConvNet on Continual ImageNet)
>
> | Algo| Task ID | FTLE < -10 | -10~0 | 0~10 | FTLE>10 |
> |-|-|-|-|-|-|
> | No Intv. | 1000 | 0| 0 | 0.10| 0.90|
> | | 4900 | 0.61| 0 | 0| 0.39|
> | L2 init | 2000| 0 | 0.11| 0.89| 0 |
> || 4500| 0 | 0.09| 0.91| 0 |
> | G-mixup | 2000| 0 | 0 | 1| 0 |
> || 4500| 0 | 0 | 1| 0 |
> | CBP | 2000| 0 | 0 | 1| 0 |
> || 4500| 0 | 0 | 1| 0 |
>
> As shown above, methods that aim to address LoP also maintain FTLE within a healthy range, albeit with different mechanisms. In contrast, baselines exhibit significantly lower(higher) FTLE values, indicating repr. collapse and expansion.
> Theoretically, we've also demonstrated with theorem 3.1.2 that extreme FTLE values characterize the collapse and over-stretch of repr. space, which would lead to LoP.
> Although it is hard to _rigorously prove_ that a system will not experience LoP at all when FTLE looks health, we consider these evidences to be enough to claim that such events are highly unlikely.
>
> **W7**
>
> We note that the smoothing effect of mixup-style augmentations align well with our theoretical framework, which highlights the importance of maintaining a well-conditioned (avoiding extreme FTLE values) repr. space during continual learning, but it could be overshadowed by the effect of sample diversity. Mixup promotes smooth transitions in the repr. space by interpolation of intra-class samples and labels, which can mitigate Type-2 LoP. However, mixup retains unchanged intra-class label, i.e, no smoothing, which limits its ability to prevent Type-1 LoP.
> Therefore, to empirically demonstrate that most of the performance improvement from G-mixup comes from mitigating LoP, rather than from the increased sample diversity, we restrict both mixup and G-mixup to intra-class sample pairs, thereby removing their `inter-class` smoothing effects and equalizing sample diversity. Whereas, G-Mixup still employs label enhancement, which systematically varies intra-class labels (Eq. 15), offering smoothing for the intra-class repr. space.
> As shown in the table below, mixup rapidly goes into type 1 LoP, whereas G-Mixup consistently mitigate type 1 LoP, supporting continual learning.
>
> | Task(x500) | 0-1    | 1-2    | 2-3    | 3-4    |
> | ---------- | ------ | ------ | ------ | ------ |
> | G-Mixup    | 0.8641 | 0.8715 | 0.8792 | 0.8778 |
> | Mixup      | 0.8635 | 0.5813 | 0.5000 | 0.5000 |
>
> **W8 & Q7**
>
> Indeed, the stability–plasticity dilemma has been a central topic in the CL community. However, our study is positioned within the emerging line of research that focuses on _plasticity loss_—a perspective that emphasizes the opposite concern: maintaining the network's ability to learn new tasks effectively.
>
> For a typical LoP benchmarks like Continual ImageNet which involves up to 5,000 tasks, backward consistency cannot be guaranteed. E.g, methods like CBP explicitly reset parts of the network at each step, fully embracing the loss of old task performance in favor of forward generalization. A related example is in reinforcement learning, where preserving value functions learned under outdated policies can hinder new learnings.
>
> That said, we agree with the reviewer that empirical measurement of stability is valuable. To this end, we have evaluated our method using the Forgetting Measure, and include the results in the following table:
>
> ConvNet on Continual ImageNet
> | Algo| FM|
> |-|-|
> | L2 init| 0.27 |
> | CBP | 0.27 |
> | G-mixup [Ours] | 0.28 |
>
> SmallConvNet on Continual ImageNet
> | Algo| FM|
> |-|-|
> | L2 init| 0.25 |
> | CBP | 0.23 |
> | G-mixup [Ours] | 0.30 |
>
>
> **Q6**
> Although our current theoretical analysis is primarily developed for binary classification tasks to ensure clarity and analytical tractability, we have empirically validated our approach on the Class-Incremental CIFAR-100 benchmark (see L299), where the number of classes increases to 100. We think that there are no fundamental obstacles to extending our framework to the multi-class classification setting.
>
> **Q8**
> We appreciate the reviewer’s emphasis on hyperparameter robustness, which is indeed a crucial aspect for practical deployment. To examine this, we conducted additional experiments using both SmallConvNet and ConvNet on Continual ImageNet, exploring the amplification factor $M \in [0.1, 0.3]$ and Beta prior $\in [0.5, 1.5]$. Results show only mild performance variation: SmallConvNet ranged from 0.875 to 0.886, and ConvNet from 0.895 to 0.904, indicating that G-Mixup is reasonably robust without fine-tuning.

---

> > ### Comment · Reviewer_FyTC · 2025-08-04
> >
> > Thank you for the rebuttal. The authors have made a commendable effort in their rebuttal, providing additional experiments that address several of the initial concerns. However, a significant question remains as to whether the approach focused solely on plasticity can be extended into a framework that also addresses catastrophic forgetting. The primary objective of CL is to perform well on both existing and new tasks. An approach that only demonstrates strong performance on new tasks is fundamentally no different from simply training a new model from scratch for each task. From a CL perspective, it would be more insightful to focus on the inherent trade-off between learning and forgetting. A direct analysis of this relationship would constitute a more valuable contribution. Alternatively, the paper would be significantly strengthened by a discussion on the potential for a framework that synergistically combines the proposed plasticity method with existing anti-forgetting techniques.
> >
> > Regarding the response to W4 & Q5 (Q4):
> >
> > The question is not about assigning different labels to identical image instances. For example, in the Continual ImageNet experiment, each image retains its original label. Instead, each task involves classifying samples from two newly selected classes. Given this setup, the question is whether the equation used by the authors remains valid under such conditions, or whether a Type-1 LoP can still arise in this scenario—even though labels themselves are not reused or reassigned.
> >
> > Regarding the response to W5, W6 & Q6:
> >
> > The FTLE distribution indicates that G-mixup effectively mitigates the LoP issue compared to other methods. However, the FTLE distribution of CBP appears quite similar to that of G-mixup, suggesting both methods achieve comparable class separation in latent space. This raises an important question: How do the authors interpret the performance advantage of G-mixup over CBP, despite their similar FTLE characteristics? Clarifying this discrepancy would help better understand the specific benefits conferred by G-mixup.

---

> > > ### Author Response · Authors · 2025-08-06
> > > **Regarding Plasticity Loss and Catastrophic Forgetting**
> > >
> > > We highly appreciate the reviewer's recognition of our efforts, along with the reviewer's prompt reply and thoughtful clarification on the unresolved issues.
> > > Due to the character limit per comment, we will reply with 3 comments, corresponding to the 3 points raised. This is part 1/3.
> > >
> > > We fully agree that CL methods should ideally address both plasticity and forgetting, and understanding the stability–plasticity trade-off is central to advancing the field.
> > > However, we'd like to clarify the objective of the _plasticity loss_ researches first. As the reviewer rightly notes, strong performance on new tasks can be achieved by training from scratch. However, in practice, this is often not viable or effective.
> > >
> > > When training from scratch for each new task, limited training data often prevents the model from achieving satisfactory performance. In contrast, a network that has been continually trained on previous tasks can reuse learned features. As shown in Table 1,2 of our main text, despite similar task difficulty from Task 1 to 5000, the performance of G-Mixup and CBP gradually improves. This suggests that previous learned feature representations positively contribute to the performance on later tasks. Alternatively, one might consider joint training on both old and new tasks. However, this approach would incur prohibitive computational and memory costs—especially in large-scale settings.
> > >
> > > Therefore, the core motivation behind LoP studies is: **Can we leverage features learned from old tasks to optimize performance on new tasks efficiently, without explicitly preserving the old tasks' performance?** i.e., like those who play a musical instrument consistently improve their skills by practicing different songs even if they forget the specific songs later on. Such scenarios are common in RL, where policies are updated continuously rendering old ones obsolete but some of the feature representations of input data and common rules are of great use when facing new environments. The same goes for the settings of continual image classification where methods that prioritize plasticity—such as G-mixup—are advantageous. They free the network from the constraints of old classification boundaries while utilizing learned features to achieve better performance.
> > >
> > > Nevertheless, knowing how much these methods impact the old tasks' performance is of interest to various CL studies. We follow the reviewer's advice and conducted additional experiments to explore the stability-plasticity trade-off, measuring stability systematically for 3 LoP mitigation methods with Average Forgetting (AF) and Relative Forgetting (RF). Upon observing that the average old tasks' performance of all methods drop to near 0.50 (chance level, binary classification) we computed the standard deviation (Std) from the chance level to help capture the degree of information retained from old tasks.
> > >
> > > **Table A. Continual ImageNet (SmallConvNet)**
> > >
> > > | Method  | AF   | RF   | Std  |
> > > | ------- | ---- | ---- | ---- |
> > > | L2 init | 0.25 | 0.32 | 0.15 |
> > > | CBP     | 0.23 | 0.29 | 0.12 |
> > > | G-Mixup | 0.30 | 0.36 | 0.13 |
> > >
> > > **Table B. Continual ImageNet (ConvNet)**
> > >
> > > | Method  | AF   | RF   | Std  |
> > > | ------- | ---- | ---- | ---- |
> > > | L2 init | 0.27 | 0.33 | 0.15 |
> > > | CBP     | 0.27 | 0.33 | 0.07 |
> > > | G-Mixup | 0.28 | 0.34 | 0.11 |
> > >
> > > Indeed, through the AF and RF measurements of CBP, L2 init vs. G-Mixup (and paired with results from Table 1 and 2 in the maintext), we observed the stability-plasticity trade-off as the reviewer has foreseen, i.e., G-Mixup suffers the strongest Forgetting while achieving the best Plasticity, especially in the scenario where type 1 LoP dominates. As AF and RF quantify the gap between the initial vs. re-evaluated (after training Task 5000) accuracy, G-Mixup's higher initial accuracy naturally leads to larger AF and RF, given that the re-evaluated accuracies of all methods drop to chance level.
> > >
> > > From Table B on networks that prone to type 2 LoP, all methods show similar AF and RF, but their behaviors differ in Std. Specifically, we find CBP to exhibit the lowest Std, suggesting that its representations are largely decorrelated from earlier tasks. This is consistent with the fact that CBP resets a part of the network only based on their current utility. Meanwhile, L2 init achieves the highest Std in both expt's, indicating that its performance largely depends on how well the representations induced by the initial weights align with the old tasks, case by case. G-mixup's performance varies moderately and sits in the middle.
> > >
> > > Importantly, we note that **G-Mixup operates without any weight manipulations or resets**, unlike CBP or L2 init. This suggests that it may be more compatible with existing anti-forgetting techniques, such as EWC or replay-based methods, which could very well opens a new direction for future research of CL (thanks to the reviewer).

---

> > > > ### Author Response · Authors · 2025-08-06
> > > > **Regarding W4 & Q5 (Q4)**
> > > >
> > > > We sincerely thank the reviewer for raising this important and subtle point, and we apologize for any confusion caused by our previous response. To ensure we have now correctly understood the question, we restate it as follows:
> > > >
> > > > > If all the learned tasks consist of newly selected classes, such that these new classes have not been trained until task \(T\), it seems that $\lambda_0^{k,k'}$ cannot be defined. Under such conditions, does the equation used by the authors remain valid, and can a Type-1 Loss of Plasticity (LoP) still arise in this scenario?
> > > >
> > > > The _short answer_ is: Yes, **Type-1 Loss of Plasticity (LoP)** can still arise under such conditions, as is the validity of our theoretical analysis. However, the interpretation of the two $k$ in the notation need to be generalized to samples (of a new class) that are close to a previous class $k$, under this setting. The essence is that the collapse of repr. space is not purely local, in fact, it impacts the repr. space globally (though locally the strongest).
> > > >
> > > > We had this situation in mind when formulating the theory, but omitted explicit discussion for clarity and simplicity in the main text. Here we provide a _detailed elaboration_ as follows:
> > > >
> > > > The evolution of the repr. space in neural networks is not a purely local phenomenon occurring only at the training samples—it is a relatively **global process**. When training on a given task with two classes, each class is represented by a finite number of samples that form a discrete approximation of the underlying input distribution. While it is well known that representations of same-class samples tend to a same representation during training, the remaining question is: **what happens to the representations of unsampled data points?** These points—including those belonging to the future tasks with previously unseen classes (as the reviewer has emphasized)—are also affected.
> > > >
> > > > In the main text, **Toy Model 1** illustrates this point. In that setup, each task uses a newly sampled dataset from a 2D space, with class labels randomly assigned per task. Sample reuse across tasks is extremely rare. Yet, despite this, we still observe clear Type-1 LoP, validating that label reuse is not a necessary condition for representational collapse.
> > > >
> > > > This phenomenon occurs because training a neural network modifies not only the representations of the observed samples but also induces global transformations across the entire input space. Specifically, an unsampled data point's representation will be shaped by the local neighborhood of trained samples. If the surrounding samples all belong to a single class, the unsampled point’s representation will tend to collapse toward that class center. If the neighborhood contains samples from multiple classes, the local repr. space may exhibit expansion due to dissimilar representations on either side.
> > > >
> > > > This mechanism underlies the generalization ability of neural networks: since test data are assumed to be drawn from the same distribution as training data, most test points lie close to training points and hence follow similar transformation dynamics. This would be immediately evident if the reviewer is familiar with Frequency Principle (Z.Q. Xu et. al., 2019), stating that DNN learns a _low-frequency_ (i.e., smooth and non-local) version of the input distribution first.
> > > >
> > > > In addition, the **Neural Collapse** literature supports this perspective. At convergence, the classification layer of a trained network effectively performs a nearest-class-center decision, and the repr. space is divided into distinct class regions. Thus, during training, intra-class regions collapse and inter-class boundaries expand—not only for observed points but also for the nearby representations.
> > > >
> > > > In our formulation, $\lambda_0^{k,k'}$ refers to the pairwise representational distance **at initialization**, even for class $k$ that is not encountered until task $T$. Despite not having been trained directly, class $k$’s latent region may have already been shaped by previous tasks due to representational deformation of nearby regions.
> > > >
> > > > As a result, when class $k$ is eventually introduced, its repr. space may already contain:
> > > >
> > > > - **Compressed regions**—lying near to previously collapsed areas of other classes (prone to Type-1 LoP).
> > > > - **Stretched regions**—lying near inter-class boundaries from previous tasks (prone to Type-2 LoP).
> > > >
> > > > Finally, one might ask if the collapse and expansion effects from different tasks could cancel out over time—i.e., a spontaneous equilibrium between the two? While theoretically possible, this balance is unstable in practice, and highly dependent on the spatio-temporal distribution of the samples. Therefore, while the opposing forces might delay the onset of LoP, it **cannot prevent it entirely**.
> > > >
> > > > We thank the reviewer again for this thoughtful question, and we will clarify this point more explicitly in the revised manuscript.

---

> > > > > ### Author Response · Authors · 2025-08-06
> > > > > **Regarding W5, W6 & Q6**
> > > > >
> > > > > We thank the reviewer for this thoughtful and insightful question. We have greatly benefited from engaging with the reviewer's comments throughout this process.
> > > > >
> > > > > As the reviewer points out, both G-Mixup and CBP exhibit relatively well-behaved FTLE distributions, indicating that they effectively avoid representational collapse or uncontrolled expansion and thereby maintain plasticity. However, the performance gap between the two methods arises not merely from their capacity to preserve plasticity, but more importantly, from the **quality of the resulted representations**.
> > > > >
> > > > > FTLE primarily reflects the degree of separation between nearby samples in the latent space and is thus a proxy for plasticity. A collapsed or overly stretched representation space generally leads to poor plasticity and reduced learnability. However, **task performance depends not only on the model’s ability to learn the current task (plasticity), but also on how well previously acquired knowledge contributes to learning the new task**.
> > > > >
> > > > > This is where G-Mixup and CBP differ. CBP maintains plasticity by frequently resetting a portion of the network’s parameters. While this helps prevent representational rigidity, it also **poses a risk of discarding useful features** acquired in earlier tasks, especially if those features are not actively reused over a few subsequent tasks.
> > > > >
> > > > > This effect is empirically observable in Tables A and B (provided in this response above): the backbone network trained with CBP exhibits the **lowest standard deviation from chance level** when evaluated on prior tasks after learning all 5000 tasks. This suggests that CBP’s representations have become largely **decorrelated from previously learned classification heads**, indicating potential forgetting of important representations due to repeated resets.
> > > > >
> > > > > In contrast, **G-Mixup does not rely on explicit weight reinitialization**. As a result, it is more likely to preserve and reuse valuable features learned from past tasks while still maintaining plasticity through a smooth latent space geometry. This facilitates a higher-quality feature representations that can generalize better across tasks and contributes to its stronger empirical performance.
> > > > >
> > > > > This motivation is central to our work: we aim to develop a theoretical understanding of LoP and, building upon that, propose a method that maintains plasticity **without relying on randomization or weight resets**—thus minimizing disruption to previously learned features (despite the penultimate-layer representations being disrupted by new classification boundaries) and achieving better continual learning performance.
> > > > >
> > > > > We will revise the manuscript to include this discussion and clarify the distinction between plasticity preservation and representational quality, which underlies the observed performance advantage of G-Mixup over CBP.

---

> > > > > > ### Comment · Reviewer_FyTC · 2025-08-06
> > > > > >
> > > > > > Thank you so much for the detailed explanation and additional experiments. I will keep the positive score.

---

> > > > > > > ### Author Response · Authors · 2025-08-09
> > > > > > >
> > > > > > > We are grateful to the reviewer for the time and effort devoted to evaluating our work thoroughly, and we have enjoyed the active discussion with reviewer on these highly insightful points regarding the details of our study, which have greatly helped us refine and strengthen the paper. We will incorporate the results stemmed from the in-depth discussion with the reviewer into the revised manuscript. Last but not the least, we highly appreciate the reviewer for keeping the positive score and feel encouraged by the recognition.

---

### Official Review · Reviewer_BXEg · 2025-07-02

**Clarity:** 2
**Significance:** 2
**Originality:** 3
**Rating:** 4
**Confidence:** 2

**Summary:**

The authors study the Loss of Plasticity (LoP) through the lens of the theory of neural collapse and finite-time Lyapunov exponents (FTLE) analysis. Their claim is that LoP can manifest in two separate regimes:
(1) Type 1 LoP is characterized by highly negative FTLEs, where the network is prevented from learning due to the collapse of representations
(2) Type-2 LoP is characterized by excessively positive FTLEs, where the network can train well but which test accuracy on new tasks degrades
The authors propose an approach called Generalized Mixup, which allows to mitigate LoP and which is inspired by these 2 regimes.

The authors conduct toy and real-datasets experiments to showcase their proposed Generalized Mixup strategy as well as to demonstrate two types of the LoP.

**Questions:**

* What is the reasoning behind architectural design choices for toy experiments for Type 1 & Type 2?

* What is the difference in the experimental regime for Figure 4.a & 4.c and Figure 4.b and 4.d?

* Could the story which is suggested by the authors be simply explained by (a) lack of network capacity (for Type 1), and by (b) overfitting (for Type 2), as opposed to their theoretical framework?

**Ethical Concerns:**

["NO or VERY MINOR ethics concerns only"]

**Final Justification:**

The authors provided a clarification for my mis-understanding of their contribution and I now acknowledge that their theoretical contribution is novel.

**Limitations:**

Please see weaknesses.

**Paper Formatting Concerns:**

No concerns

**Quality:**

3

**Strengths And Weaknesses:**

Strengths:
* A novel perspective on the LoP through the theory of Neural Collapse and finite-time Lyapunov exponents (FTLE) analysis. Their theory identifies two separate regimes of the LoP.
* A new empirical method for augmenting data, Generalized Mixup, which can tackle the LoP. I think it is particularly interesting because generally LoP approaches do not focus on data augmentations
* Experimental results showcasing that Generalized Mixup could help with the LoP
* (Controversial) Toy experiments showcasing two regimes of LoP through the lens of the proposed theory

Weaknesses:
* In my opinion, theoretical contribution is a bit limited. The proposed approach feels more like a "diagnostic" of what is happening, rather than a theory which explains exactly why the LoP happens. My opinion is based on Figure 4, where in both Low dim and High Dim spaces, there are both Type 1 and Type 2 regimes happening. Why is this the case? When I check the toy experiments, it looks like Type 1 happens when the network capacity is low (because authors use 10 hidden units per layer) while Type 2 happens when the network capacity is high (because they use 300 hidden units). Based on this, I would have assumed that In figure 4, in low dim space, only Type 1 would happen, but there is Figure 4,c where both happen. Similarly for Figure 4,b and d. Therefore, it feels that this perspective on Type 1 and Type 2, is something which we can look a-posteriori as a diagnostic of what went wrong, but not as something which can be used to predict LoP.

* The authors don't really explain in the main text why they use 10 hidden units per layer for Type 1 toy experiment and 300 hidden units per layer for Type 2 toy experiment. It feels arbitrary.

* Same criticism, for Figure 4, the authors don't explain the difference between (a) and (c); and (b) and (d).  It's unclear why we have Type 1 and then suddenly Type 1+ Type 2.

* Based on weakness I mentioned above, could the story which is suggested by the authors be simply explained by (a) lack of network capacity (for Type 1), and by (b) overfitting (for Type 2), as opposed to their theoretical framework.

* Based on these, I think the significance of the work is a bit limited.

* I have no concerns regarding the Generalized mixup, this looks like something which helps for addressing LoP.

---

> ### Author Rebuttal · Authors · 2025-07-31
>
> We sincerely thank the reviewer for the constructive feedbacks. We would like to start by an overall reply to provide general clarifications about the issues raised here, given that some of the central idea of our work might have not come through. Then, we will follow with responses to the specific concerns.
>
> ### Overall reply
>
> This paper uncovered the fundamental mechanisms underlying LoP in continual learning (CL), i.e., the compressing/stretching of the representational(repr.) space. This primary contribution is a theoretically grounded diagnosis of the emergence of two types of LoP though a-posteriori. It is based on how CL compress and over-stretch the repr. space by repeated exposure to overlapped input distribution from intra-class and inter-class samples, respectively (as proved in theorem 3.1.2). The dynamics of the manipulation of repr. space geometry is characterized by FTLE. Recently, LoP has been associated with various observations, such as gradient vanishing or an increase in dead neurons. However, as pointed out in Lyle et al., (2024), these singular phenomena are insufficient to fully explain LoP on their own. This points to the lack of unified theoretical understanding of LoP since those phenomena are not the direct causes. Despite this, numerous methods have been proposed to mitigate LoP. Their strategies can broadly be classified into two categories, as the following.
>
> One approach targets specific manifestations of LoP, such as large weight norms. For instance, Dohare et al. (2024) investigated the use of L2 regularization to constrain weight norms and found that it can partially mitigate LoP. Our theoretical framework provides a deeper understanding of how these measures work: the network used in their study is relatively wide and hence susceptible to Type 2 LoP. In such cases, L2 regularization prevents excessively large weights, thereby mitigating Type 2 LoP. However, in networks with initially lower repr. dimensions (narrow networks), which are more prone to Type 1 LoP, L2 regularization can actually impair performance. (We want to emphasize here that the size of repr. dimensionality does not equal to the network capacity, as initially large/small repr. space can still be easily stretched or compressed extremely during CL.)
>
> Another approach builds on the observation that plasticity is highest at initialization. Techniques such as continual weight re-init (e.g., CBP) or regularization toward initial weights (e.g., L2-init) aim to keep the network in a more plastic regime. While effective in alleviating LoP, these methods suffer from several drawbacks: weight re-init can erase previous learning, while regularization toward initial weights can constrain the network's expressivity. Our work, in contrast, mitigate LoP based on the theoretical understanding that the plasticity can come from maintaining a reasonably sized (for Type-1 to not collapse) and smoothed (for Type-2 to not become chaotic) repr. space without going back to the initialized state, thus avoiding the drawbacks.
>
> (relates to )While we appreciate the reviewer’s insightful attempt to relate Type 1 and Type 2 LoP to limited capacity and overfitting respectively, we respectfully note that this interpretation does not fully capture the mechanisms we aim to describe. This is precisely why we introduced the FTLE framework to characterize the dynamics of repr. space rather than relying on conventional notions of capacity or overfitting. We appreciate the chance to reiterate the essence of our work here.
>
> Our theory demonstrates how repr. geometry crucially drive the network dynamics into two regimes. On the higher level, the dimensionality of the repr. space governs the general direction of the dynamics, i.e., as the dimensionality (which can be set by network width only `initially`) increases, the collapse of intra-class space (related to type-1 LoP) weakens and the chaotic expansion of inter-class space (related to type-2 LoP) dominates. On the lower level, only the local geometry can determines , effectively measured by FTLE. In addition to the theory described in the main text, we support this claim with empirical evidence shown in the following table:
>
> In the following two tables, we report the FTLE stats for networks of varying widths. In **Table A1**, input samples are uniformly distributed, conformed to our theoretical assumption. As network width increases, the minimum FTLE quickly approaches zero much more quickly than its imbalanced counterpart.
>
> However, when the input distribution is highly imbalanced—as shown in **Table A2**, where one class dominates the majority of the space—the intra-class representation vectors become less orthogonal and more prone to collapse, demonstrated by more negative min FTLE vs. the ones for uniform input distribution.
>
> - **Table A1: FTLE statistics under uniform input distribution**
>
> | HiddenDim | Max    | Min      | Avg     |
> | --------- | ------ | -------- | ------- |
> | 10        | 7.6742 | -13.0385 | -3.9194 |
> | 150       | 9.0213 | -3.3198  | 1.4912  |
> | 300       | 9.2577 | -2.4663  | 2.2920  |
>
> - **Table A2: FTLE statistics under class-imbalanced input distribution**
>
> | HiddenDim | Max     | Min      | Avg     |
> | --------- | ------- | -------- | ------- |
> | 10        | 8.8215  | -14.0616 | -4.5067 |
> | 150       | 9.9708  | -5.2987  | -0.0250 |
> | 300       | 10.1752 | -5.1530  | 0.6825  |
>
> Beyond this simplified toy model setting, data distributions are structured, various activation functions and optimizer introduces additional constraints in real training scenarios. These factors modulate the global and local dynamics of the collapse and expansion of the repr. space, but do not alter the general prediction from our theory:
>
> - In low-dim spaces, repr. space collapses (Type 1 LoP), though expansion can still occur.
> - In high-dim spaces, repr. space expands chaotically (Type 2 LoP), though collapse may still be present.
>
> Our empirical observations also align with this: we never observed pure Type 2 LoP in low-dimensional networks, nor pure Type 1 LoP in high-dimensional ones. In Figure 4 of the main text (detailed experiment settings can be found in Appendix E), we chose two widely used activation functions—ReLU (scale-invariant) and tanh (saturating)—to illustrate all four theoretically predicted regimes. One can observe that the narrower ReLU networks in Figure 4C exhibit significantly faster onset of Type 1 LoP than the wider network in Figure 4D, again consistent with our theoretical expectations.
>
> Therefore, Type 1 and Type 2 LoP are the emergent behavior driven by the evolving geometry of the repr. space during CL:
>
> - Type-1 LoP arises from repr. collapse. While narrower networks are more susceptible to such collapse, wider networks tend to collapse more slowly, though they may still exhibit collapse depending on various factors.
> - Network capacity alone cannot explain Type 1 LoP. Increasing capacity (width) may slow down collapse but cannot eliminate it.
>
> Regarding overfitting, it typically refers to excessively tailoring a model to a specific dataset, impairing generalization. While overfitting can account for some aspects of Type 2 LoP, it is not the primary cause in our CL setup. In our setting, the model learns 5000 tasks of comparable difficulty, and performance degradation emerges gradually, unlike the abrupt generalization loss associated with overfitting.
>
> Generalization in DNN is supported by mapping similar inputs to similar representations. At initialization, FTLE values are approximately zero, indicating distance-preserving mappings conducive to generalization. However, as training proceeds, high-FTLE boundaries proliferate the repr. space, stretching and distorting these local mappings and degrading generalization. This happens regardless even when the learnt boundaries are perfectly generalizable for standalone tasks. Therefore, in contrast to overfitting—which is characterized by overly complex boundaries—Type 2 LoP reflects widespread instability caused by the aggregation of the stretching effect by more and more learnt boundaries (regardless of their generalizability) that finally distort the repr. space in various direction causing chaotic behavior.
>
> We deeply appreciate the opportunity to engage in this discussion. The reviewer’s thoughtful comments helped sharpen our own understanding, which we will integrate into the final manuscript to improve the clarity and impact of the work.
>
> ### Regarding specific concerns
>
> - **W1**: With the overall reply above, we'd like to argue that the provided theory offer much more than a-posteriori diagnosis of LoP, and it reaches the direct cause of the two types of LoP, which also prompt the design of G-mixup that mitigate LoP in advance.
>
> - **W2 & Q1:** Our core motivation is to disentangle the mechanisms behind LoP. We deliberately selected toy models that isolate Type 1 and Type 2 LoP. Therefore, we used a narrower network with 10 hidden units which is prone to Type 1 LoP vs. a wider network of 300 hidden units which is prone to Type 2 LoP. These choices of setups serve as archetypes to illustrate the distinct underlying mechanisms. Networks of all widths follow our theory; what differs is the dominance and the onset speed of each type of LoP.
>
> - **Q2 & W1 & 3:** As discussed, collapse/expansion dynamics depend on activation function, data distribution, and training details. Figure 4 employs ReLU (Figure 4a,d) and tanh (Figure 4b,c) to depict typical real-world cases where both types can co-occur.
> - **Q3 & W4:** While on the surface-level similarities exist between Type 1/2 LoP and capacity/overfitting, the former reflect dynamic changes in representation during CL, instead static model properties that governs single task learning.
>
> We are willing to further clarify during discussion.

---

> > ### Comment · Reviewer_BXEg · 2025-08-08
> > **Response**
> >
> > I would like to thank the authors for their reply. It addresses my concerns and clarifies their contributions. I am increasing my
> > score.
> >
> > I have a few suggestions.
> >
> > It would be clearer if you specify in the main text (or in a figure caption) that Figure 4 (a), (c) use ReLU activation, (b), (d) use Tanh.
> >
> > > Regarding overfitting, it typically refers to excessively tailoring a model to a specific dataset, impairing generalization. While overfitting can account for some aspects of Type 2 LoP, it is not the primary cause in our CL setup. In our setting, the model learns 5000 tasks of comparable difficulty, and performance degradation emerges gradually, unlike the abrupt generalization loss associated with overfitting. ... Generalization in DNN is supported by mapping similar inputs to similar representations. At initialization, FTLE values are approximately zero, indicating distance-preserving mappings conducive to generalization. However, as training proceeds, high-FTLE boundaries proliferate the repr. space, stretching and distorting these local mappings and degrading generalization. This happens regardless even when the learnt boundaries are perfectly generalizable for standalone tasks. Therefore, in contrast to overfitting—which is characterized by overly complex boundaries—Type 2 LoP reflects widespread instability caused by the aggregation of the stretching effect by more and more learnt boundaries (regardless of their generalizability) that finally distort the repr. space in various direction causing chaotic behavior.
> >
> > I think it would be worthwhile to include a discussion of connection / difference of Type 2 LoP versus overfitting.

---

> > > ### Author Response · Authors · 2025-08-09
> > >
> > > We will follow the reviewer’s suggestions by explicitly indicating, in both the main text and the figure captions, the types of activation functions used in the different subplots, thereby making the presentation clearer for readers. In addition, as suggested, we will add a dedicated discussion in the revised manuscript on the relationship and distinction between Type 2 LoP and overfitting. Incorporating the reviewer’s thoughtful comments and insightful suggestions both from the initial review and during the discussion period into the manuscript will greatly improve the overall quality, clarity and the impact of our work, for which we cannot be more grateful.
> > >
> > > Finally, we'd like to thank the reviewer for taking the time and effort to review our work, and for reconsidering its contributions based on our rebuttal. We highly appreciate the reviewer’s final recognition.

---

### Note · Authors · 2025-08-13

We thank all the reviewers, ACs and SACs for your time and effort invested in reviewing our work. The constructive discussions with the reviewers have allowed us to clarify key points, incorporate valuable suggestions, and strengthen our study. We are also deeply grateful for the reviewers’ high recognition of the significance and contributions of our work. Based on the reviewers' final replies, we believe that all major concerns have been effectively addressed, which will be reflected in the revised manuscript. In the following, we briefly summarize our contributions and key points from discussions with the reviewers.

LoP is a key challenge in CL, which manifests as a progressive decline in a network’s ability to adapt to new tasks under non-stationary data. Existing explanations for LoP (e.g., dead/dormant neurons) remain debated without a unifying theory. We combine FTLE analysis with the UFM framework to show—for the first time—that LoP comprises two types: Type 1, representational (repr.) collapse, poor train/test performance; Type 2, chaotic repr. dynamics, good train but poor test performance, confirmed by toy and real-world datasets. Based on these novel and first principle-based understandings, we propose G-Mixup, which mitigates LoP (both types) by smoothing the repr. space (for both intra-class and inter-class), i.e., addressing the direct cause of LoP, thus outperforms prior empirical-based approaches that constrain weights magnitude or partially reset the network near initialization.

With Reviewer BXEg, we reached a shared understanding that the causes of the LoP are distinct from "lack of network capacity" or "overfitting", underscoring the novelty of our findings. Under reviewer FyTC and GUee's encouragement,  we examined the stability–plasticity trade-off, which is related to LoP but conceptually distinct: the former targets optimal performance across all tasks in CL, whereas LoP focuses on current-task performance. This discussion clarified our scope and suggested that combining stability-preserving methods with G-Mixup could be a promising direction. Reviewer m8rC’s suggestion to directly regularize FTLE for LoP mitigation helped us further validated and strengthened our theory (using a toy dataset, computing FTLE on real datasets is still costly). All these discussions has raised the potential of our paper to impact future researches in CL.

We hope that AC and the reviewers will find our work a valuable contribution to NeurIPS 2025.

---

### Decision · Program_Chairs · 2025-09-17

**Decision:**

Accept (poster)

**Comment:**

This paper offers a unique perspective on loss of plasticity, with clarity and effective use of visualizations. Notably, G-Mixup is derived using theoretical analysis rather than being an ad-hoc solution. However, reviewers emphasized the need for more comprehensive comparisons and questioned whether the FTLE-based theory truly explains loss of plasticity beyond being a diagnostic tool. The authors are encouraged to address these concerns in the camera-ready version.